# Evaluation of ERA5 and Dynamical Downscaling for Surface Energy Balance Modeling at Mountain Glaciers in Western Canada

Christina Draeger, Valentina Radić, Rachel H. White, and Mekdes Ayalew Tessema

Department of Earth Ocean and Atmospheric Sciences (EOAS), The University of British Columbia, Vancouver, Canada

**Correspondence:** Christina Draeger (cdraeger@eoas.ubc.ca)

## Abstract

Regional-scale surface energy balance (SEB) models of glacier melt require forcing by coarse-gridded data from reanalysis or global climate models that need to be downscaled to glacier scale. As on-glacier meteorological observations are rare, it generally remains unknown how exact the reanalysis and downscaled data are for local-scale SEB modeling. We address this question by evaluating the performance of reanalysis from the European Centre for Medium-Range Weather Forecasts (ERA5 and ERA5-Land reanalysis), with and without downscaling, at four glaciers in Western Canada with available on-glacier meteorological measurements collected over different summer seasons. We dynamically downscale ERA5 with the Weather Research and Forecasting (WRF) model at 3.3 km and 1.1 km grid spacing. We find that our SEB model, forced separately with the observations and the two reanalyses, yields <10 % difference in simulated total melt energy and shows strong correlations (0.86) in simulated timeseries of daily melt energy at each site. The good performance of the reanalysis-derived melt energy is partly due to cancellation of biases between overestimated incoming shortwave radiation and substantially underestimated wind speed and subsequently turbulent heat fluxes. Downscaling with WRF improves the simulation of wind speed, while other meteorological variables show similar performance to ERA5 without downscaling. The choice of WRF physics parameterization schemes is shown to have a relatively large impact on the simulations of SEB components, but a smaller impact on the modeled total melt energy. The results increase our confidence in dynamical downscaling with WRF for long-term glacier melt modeling in this region.

**Keywords:** Mountain glaciers, surface energy balance, dynamical downscaling, WRF model, high-resolution atmospheric modeling, melt modeling

## 1 Introduction

In Western Canada, streamflow from glacier runoff during hot and dry seasons is essential for water supply, hydropower generation and agricultural irrigation (Schindler and Donahue, 2006; Anderson and Radić, 2020). Over the last several decades, glaciers in this region have already lost and are increasingly losing a considerable amount of mass (Larsen et al., 2007; Arendt et al., 2009; Zemp et al., 2015; Hugonnet et al., 2021). This trend of glacier retreat will continue as Western Canada is expected to see unprecedented changes to glacierized watersheds, with a predicted loss of more than 70 % of current glacier ice volume

by 2100 (Clarke et al., 2015; Rounce et al., 2023). The regional projections, however, still carry a large amount of uncertainty, especially at the local scale of individual glacierized watersheds where the impact of glacier retreat on freshwater resources will be the most consequential (Anderson and Radić, 2020). One of the main sources of this uncertainty comes from the use of empirical models of glacier melt, commonly known as temperature-index models. While these models are relatively simple to implement in the regional and global assessments of glacier mass balance, they are heavily reliant on calibration and on temperature as the sole driving variable of glacier melt (Hock, 2005). Because of their high sensitivity to calibration parameters and to downscaled temperature data from reanalysis or global climate models, the mismatch between modeled and observed seasonal melt rates of individual glaciers can exceed 50 % (e.g., Radić et al., 2014; Clarke et al., 2015). While a substantial progress has been made over the last decade to advance the ice dynamics modeling by transitioning from empirical towards physics-based approaches (e.g., Rounce et al., 2023), melt modeling of glaciers in Western Canada, as well as worldwide, still heavily relies on empirical approaches.

By contrast to the empirical models, physics-based models of glacier melt account for all components in the surface energy balance (SEB) that affect surface melt. Since these models capture the physical processes that are happening at the glacier surface, they do not rely on the temporal stationarity of melt factors, as is the case in temperature-index models. However, they require a larger number of input variables, including incoming shortwave and longwave radiation, temperature, relative humidity, wind speed and precipitation. SEB models of various complexity have been applied to individual glaciers worldwide, including several glaciers in Western Canada (e.g., Ebrahimi and Marshall, 2016; Fitzpatrick et al., 2017; Marshall and Miller, 2020; Kinnard et al., 2022), showing good resemblance between modeled and observed melt as long as the SEB models are forced by on-glacier meteorological observations. The caveat with these models, however, is that the on-glacier measurements of all SEB components are sparse in space (fewer than 100 sites worldwide, and only a handful in Western Canada) and of short duration (over one or two melt seasons on average). Thus, to produce long-term simulations of glacier melt and mass balance at regional scales, the input to SEB models needs to come from readily available climate reanalysis datasets or global climate models (GCMs). Since their native output is provided on a spatial grid too coarse to adequately resolve key processes contributing to local-scale melt, the scale mismatch if often addressed through statistical and, to much lesser extent, dynamical downscaling.

Due to its simplicity, statistical downscaling (e.g., correcting temperature with elevation using an atmospheric lapse rate) is a more popular technique than the computationally expensive dynamical downscaling, i.e. running a high-resolution regional climate model. Nevertheless, as statistical downscaling relies on simplified assumptions (e.g., the existence of linear relationships between local and large-scale climate variables), the technique introduces another source of error or uncertainty into the model output (Marzeion et al., 2020). The coarse spatial resolution of reanalysis and GCMs, as well as the limitations with statistical downscaling, led to a relatively poor performance of SEB models in the few existing modeling studies applied on regional and global scale (e.g., Noël et al., 2017; Shannon et al., 2019). An alternative to statistical is dynamical downscaling: a physics-based approach that utilizes a regional climate model (RCM), nested within a reanalysis or global climate model, to compute meteorological fields at a desirable spatial resolution, often <10 km. A well-configured high-resolution RCM, for example, can outperform radar and satellite-derived estimates of total annual rain and snowfall within mountainous regions

(Lundquist et al., 2019). While dynamical downscaling does not rely on on-glacier meteorological observations as is the case with the statistical downscaling, these observations are still critical for the evaluation of dynamically downscaled fields.

Over the last few decades, a commonly used RCM for a broad range of downscaling applications has been the Weather Research and Forecasting (WRF) model, an open-source and continuously upgraded mesoscale numerical weather prediction model (Skamarock and Klemp, 2008). To date, however, relatively few studies have evaluated the use of WRF for SEB simulations of glacier melt, and to our knowledge, no evaluation was performed for glaciers in Western Canada. A challenge in using WRF for glacier studies is the lack of on-glacier meteorological observations needed to evaluate the downscaled variables and the high computational cost in running WRF to obtain long-term climate simulations. Out of the existing studies, only few used on-glacier station data (e.g., Mölg and Kaser, 2011; Claremar et al., 2012; Eidhammer et al., 2021), while others relied on the observations from weather stations in the glacier vicinity (e.g., Collier et al., 2013, 2015). One of the first applications of WRF in glacier studies has simulated two months of SEB and mass balance at a glacier on Mt. Kilimanjaro (Mölg and Kaser, 2011). Their downscaled fields at an hourly time step and at around 0.8 km grid spacing showed strong correlation with on-glacier hourly meteorological observations. The successful WRF performance was not corroborated, however, at the coarser 3 km grid spacing. Another study applied WRF with a nesting scheme of 24 km (original domain), 8 km (nested domain), and 2.7 km (innermost nested domain) grid spacing to simulate a two-year surface mass balance for three glaciers in Svalbard (Claremar et al., 2012). Strong correlations between the output at 2.7 km grid spacing and the on-glacier observations were obtained for most downscaled variables except for the near-surface wind speed. Collier et al. (2013, 2015) developed a high-resolution interactive model at a glacier-atmosphere interface and applied it to several glaciers in Karakoram over two melt seasons. The model used three WRF domains (33 km, 11 km and 2.2 km grid spacing) and was directly coupled with the incorporated SEB model, allowing for the feedback mechanism at the glacier-atmosphere interface. Although on-glacier observations were not available for the model evaluation, the downscaled near-surface air temperature and wind speed at 2.2 km grid spacing agreed well with observations from the glaciers' vicinity, while poor performance was found for incoming shortwave radiation and precipitation (Collier et al., 2013). More recently, Eidhammer et al. (2021) used WRF downscaling to 1 km grid spacing coupled with snowpack modeling through the WRF-Hydro model (Gochis et al., 2020), showing a good agreement between the WRF output and in-situ meteorological observations at a glacier in Norway over four years.

The relatively short periods (from a few months to several years) of WRF simulations in the aforementioned glacier studies highlight the high computational cost of dynamical downscaling to a fine ($\sim 1$ km) spatial resolution. While there are studies at glacierized and mountainous terrain that used a sub-kilometer ($\sim 100$ m) grid spacing in WRF, their fine-resolution simulations were produced for only a handful of selected days (e.g., Gerber et al., 2018; Goger et al., 2022). Considering the high computational cost, the finest used spatial resolution in WRF for downscaling long-term climate simulations over a region has been on the order of a kilometer (e.g., Erler et al., 2015; Annor et al., 2018; Li et al., 2019). Therefore, when incorporating WRF into long-term glacier evolution modeling at regional scales, downscaling to a grid spacing of approximately one kilometer seems to be the computationally optimal target.

A relatively underexplored limitation in using WRF in glacier studies is the model's potentially large sensitivity to the choice of physics parameterization schemes as noted in many non-glacier studies (e.g., Liu et al., 2011; Zeyaeyan et al., 2017; Gbode

et al., 2019; Pervin and Gan, 2020; Shirai et al., 2022). When deciding on a WRF configuration for a given application, users can choose among different parameterization schemes in each category, including those for radiation, cumulus convection, microphysics, planetary boundary and surface layer. The WRF model is not only sensitive to the choice of parameterization schemes in each physics category, but also to their combination across different categories (Jung and Lin, 2016). Since it is computationally expensive to run all possible combinations of parameterizations in order to determine an optimally performing configuration, a common practice is to adopt the same or similar WRF configuration as used in previous applications. While the WRF sensitivity to the choice of physics parameterisations has been explored extensively in studies on climate dynamics and related disciplines, to our knowledge no systematic sensitivity analysis has been conducted for glacier melt modeling.

A majority of aforementioned glacier studies with WRF have downscaled the climate fields from European Centre for Medium-Range Weather Forecasts (ECMWF) Re-Analysis Interim (ERA-I; Dee et al., 2011). Relatively recently, ECMWF released ERA5 (Hersbach et al., 2020), the ERA-I successor with an enhanced modeling and data assimilation framework that utilizes a larger amount of improved observations compared to ERA-I. ERA5 data are also provided at a denser grid (30 km versus 80 km) and shorter time step (hourly versus 3-hourly) than its predecessor. As part of the ERA5 framework, ERA5-Land reanalysis was created at even denser grid (9 km) by forcing the land component of the ERA5 (Muñoz Sabater et al., 2021). Since the releases of both ERA5 and ERA5-Land, several studies have successfully applied these datasets for mass balance modeling of individual glaciers, inducing several glaciers in Central and High Mountain Asia (e.g., Azam and Srivastava, 2020; Arndt et al., 2021; Srivastava and Azam, 2022; Kronenberg et al., 2022) and two glaciers in Western Canada (Mukherjee et al., 2022). To evaluate the data from ERA5-Land, Mukherjee et al. (2022) used meteorological observations from a range of sources, but none of them included observations from their study glaciers. Despite the fine spatial resolution and improved performance of ERA5 relative to its predecessor, it remains unknown how well the reanalysis resolves key input variables for SEB modeling at glaciers in Western Canada.

Our ultimate goal is to develop a regional glaciation model, with incorporated SEB component for melt modeling, in order to project a long-term glacier evolution across Western Canada. This study, as a first step toward this goal, aims to close several identified knowledge gaps by addressing the following three questions:

(1) How well can ERA5 and ERA5-Land resolve the key input variables for SEB modeling at glaciers in Western Canada?

(2) For the SEB modeling at these glaciers, how well can WRF downscale the ERA5 reanalysis to a grid spacing of several kilometres?

(3) How sensitive are the downscaled variables to the choice of WRF parameterization schemes, and is there one most optimally performing set of parameterization schemes for our WRF application?

To address these questions, we will make use of our multi-summer and multi-station meteorological observations at four glaciers in Western Canada: three in the interior of British Columba, and one in the Yukon. The downscaling with WRF will be performed to 3.3 km and 1.1 km grid spacing using a set of different physics parameterization schemes in order to determine the "optimal" schemes for our research objectives. In the sections that follow, we start by introducing the study sites and

**Table 1.** Characteristics of the study sites. Only days with 24-hour observations have been taken into account for the observational periods.

| Glacier | Elevation range (m) | AWS coordinates (lat, lon) | Observation period |
| --- | --- | --- | --- |
| Castle Creek | 1900–2800 | 53.0508°, −120.4443° | 23 Aug – 15 Sep 2012 |
| Nordic | 2000–2900 | 51.4343°, −117.6997° | 13 Jul – 27 Aug 2014 |
| Conrad | 1800–3200 | 50.8249° , −116.9225° | 18 Jul – 05 Sep 2015 |
| | | $AWS_1$: 50.8233°, −116.9199° | 20 Jun – 24 Aug 2016 |
| | | $AWS_2$: 50.7822°, −116.9120° | 20 Jun – 24 Aug 2016 |
| Kaskawulsh | 760–2580 | 60.7589°, −139.1246° | 01 Jul – 26 Aug 2019 |

meteorological observations, followed by the description of the WRF configuration and the SEB model. We then describe the evaluation analysis and sensitivity tests used to determine the optimal WRF parameterization schemes. The paper is finalized with the presentation of results, discussion and conclusions.

## 2  Data and methods

In this section we present the observations collected from automatic weather stations (AWS) at our four study glaciers, which are used as a reference dataset for the evaluation of ERA5 and ERA5-Land reanalysis data, as well as WRF-downscaled data. We then describe the setup of the WRF model, including the choice of parameterization schemes to be used in the sensitivity tests. The AWS data, as well as the reanalysis and WRF output, are used to force a simple SEB model to simulate daily timeseries of surface energy available for melt over the observational period at the study glaciers. We briefly describe the SEB model and introduce evaluation metrics used to investigate the optimal configurations with the physics parameterization schemes in the WRF model.

### 2.1  Field sites and measurements

On-glacier meteorological measurements, as part of different research projects over the last decade, have been collected from three glaciers in the Interior Mountains of British Columbia and one large glacier in the St. Elias Mountains in the Yukon (Figure 1). The AWSs intermittently recorded data for glacier sites within different summer seasons between 2012 and 2019 (Table 1). Five AWSs recorded local meteorological variables and energy and mass fluxes in ablation zones (Castle Creek glacier, 2012; Nordic glacier, 2014; Conrad glacier, 2015, 2016; Kaskawulsh glacier, 2019), while one AWS was set up in the accumulation zone of the Conrad glacier in 2016. Topographic maps of these glaciers with the AWS locations are shown in Figure 2.

All AWSs measured the following variables: incoming and outgoing components of shortwave and longwave radiation fluxes, 2 m air temperature and humidity, atmospheric pressure, 2 m wind speed and direction, liquid precipitation, temperature

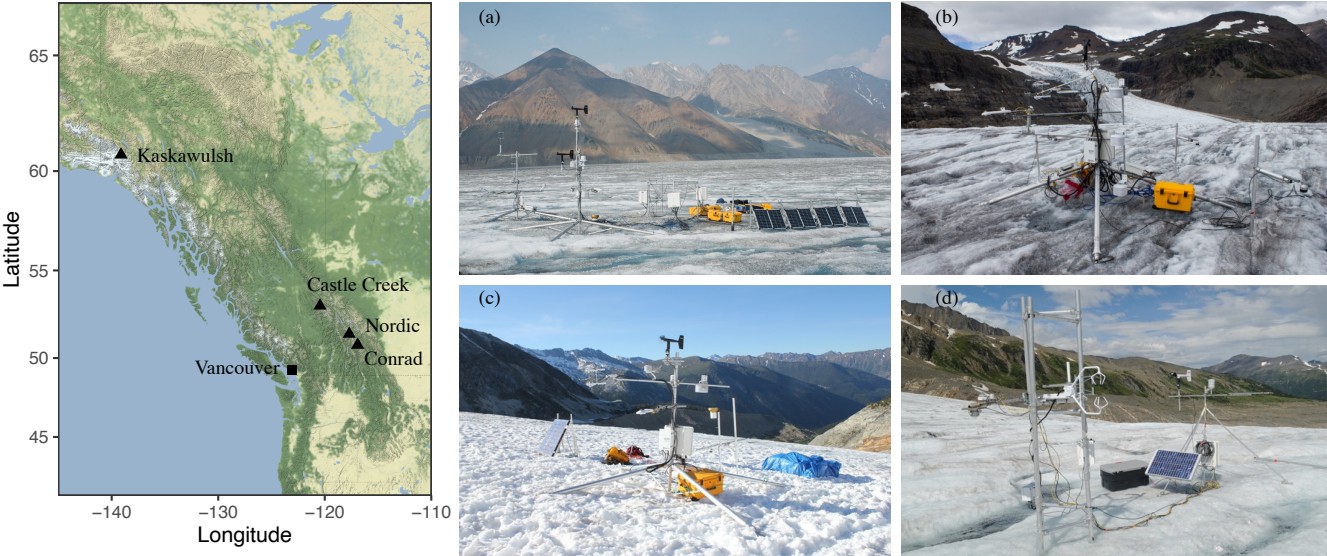

**Figure 1.** Map of Western Canada with geographic locations of the four study glaciers (black triangles on map), as well as photographs of the station setup for (a) Kaskawulsh glacier in 2019 (photo by Cole Lord-May), (b) Conrad glacier in 2016 (photo by Noel Fitzpatrick), (c) Nordic glacier in 2014 (photo by Noel Fitzpatrick), and (d) Castle Creek glacier in 2012 (photo by Valentina Radić).

in the surface layer from the surface down to 4 m depth, and surface height changes as an indicator of solid precipitation and
ablation at the site. In addition to these observations, high-frequency (20 Hz) measurements of wind speed, air temperature and humidity were collected by sonic anemometers and gas analyzers to assess the turbulent heat fluxes through the eddy-covariance (EC) method. More details on the specifications of the sensors and accuracy control at each site are given in Radić et al. (2017), Fitzpatrick et al. (2017, 2019) and Lord-May and Radić (2023). The meteorological sensors at all the sites except Castle Creek glacier were housed on a four-legged quadpod, which provided a stable platform (where any tilt was monitored
by an inclinometer) that lowered as the ice melted, and maintained a nearly constant height of the sensors above the surface (Figure 1). All variables were saved as one-minute averages except for rainfall, which was saved as one-minute totals, while the EC-derived turbulent fluxes were calculated as 30-min averages. A time-lapse camera in close proximity (∼30 m) to each AWS was used for a visual record of surface and atmospheric conditions during the observational period.

Castle Creek glacier is located in the Cariboo Mountains and contributes meltwater to Castle Creek, a tributary of the Fraser
River. The AWS on Castle Creek glacier operated at the lower part of the glacier, which was gently sloping with an approximate mean gradient of 7° (Figure 2). A melting ice surface was present during the observational period of 24 days in 2012, with some intermittent fresh snowfall events (Radić et al., 2017). Nordic and Conrad glaciers lie in the Purcell Mountains in Eastern British Columbia and are located within the Columbia River basin. The surface slope at the location of the AWS on Nordic glacier was 13°, while 8° were observed for Conrad glacier in the ablation area (AWS$_1$) and 3° in the accumulation area
(AWS$_2$). Over the course of 46 days in 2014 at the Nordic glacier site, a transitional snow surface was present for the first

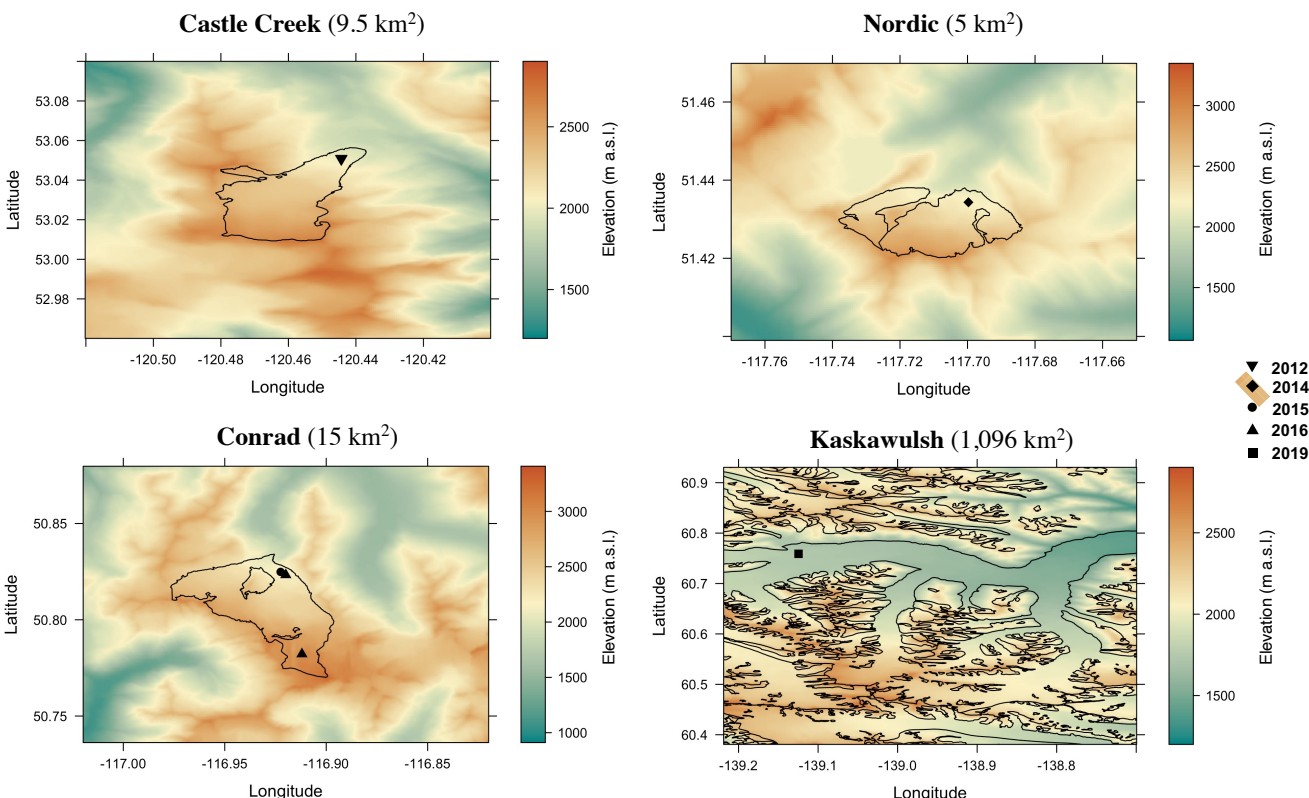

**Figure 2.** Topography maps with total glacier area (in brackets) of the domain containing the study glaciers, including the outline of each glacier from Randolph Glacier Inventory (RGI V6; RGI Consortium, 2017). The map with the Kaskawulsh glacier, showing the bulk of the glacier's ablation area, also illustrates the outlines of other smaller glaciers in the region. Different markers on the map (diamond, triangle, circle, etc.), corresponding to different years of observations, are locations of the AWSs. The topography maps were created from USGS GMTED2010 digital elevation model (DEM; EROS, 2017).

four days, with partial snow cover diminishing to a fully bare ice surface (Fitzpatrick et al., 2017). A melting ice surface was present during observations for Conrad glacier in 2015, and for most of the observational period in 2016 at the AWS in the ablation zone (Fitzpatrick et al., 2019). At the AWS in the accumulation zone of Conrad glacier, a snow surface was present throughout the observational period in 2016, and for the first ten days at the AWS in the ablation zone (Fitzpatrick et al., 2019).

Kaskawulsh glacier is located in the St. Elias Mountains and is part of the Kluane Icefield. The surface slope at the location of the AWS in the summer of 2019 was less than 2°. Throughout the observational period, the ice surface was at the melting point (Lord-May and Radić, 2023).

## 2.2 Reanalysis data

Global climate reanalysis products combine modeled data with observations from across the world to provide a globally
complete and consistent dataset of multiple climate variables of the recent past. ERA5 reanalysis provides hourly estimates of
a large number of atmospheric, land and ocean surface variables from 1950 to present at a horizontal grid spacing of 30 km for
the surface as well as 37 pressure levels from 1 hPa (top level) to 1000 hPa (bottom level; Hersbach et al., 2020). ERA5-Land
provides only surface variables on land at the interpolated 9 km grid spacing (Muñoz Sabater et al., 2021). These data are a
refinement of the land component of the ERA5 reanalysis with a higher spatial resolution, forced by meteorological fields from
ERA5. We use mainly the surface variables (details given below) from hourly ERA5 and ERA5-Land reanalysis. Hourly two-
and three-dimensional ERA5 reanalysis data is also used to provide initial and lateral boundary conditions to the WRF model.

## 2.3 WRF setup and parameterization schemes

The Advanced Research WRF (ARW) dynamics solver is a non-hydrostatic atmospheric model with fully compressible Euler
equations, solved on an Arakawa C-grid stagger in the horizontal and a terrain-following hydrostatic pressure coordinate in
the vertical (Skamarock et al., 2019). The model uses a time-split integration using a third-order Runge-Kutta scheme with
a smaller time step for acoustic and gravity-waves modes (Skamarock et al., 2019). We ran the WRF model, version 4.1.3,
configured with four nested domains of 30 km ($d_1$), 10 km ($d_2$), 3.3 km ($d_3$) and 1.1 km ($d_4$) horizontal grid spacing, with the
parent domain ($d_1$) covering the bulk of North America and the North-East section of the Pacific Ocean (Figure 3). The domains
$d_1$ and $d_2$ are kept the same for the three glaciers in the interior of British Columbia (Castle Creek, Nordic and Conrad), while
$d_3$ and $d_4$ are set differently for each of the three glaciers in order to be centered at the AWS location. We use a one-way nesting
approach, where WRF is first run for the outer domain, and then iteratively fed into the nested domains as lateral boundary
conditions. Since the outer domain has a larger grid spacing and time step than the nested domains, interpolations in both
space and time are required. This process is repeated for each pair of nested domains. WRF is initiated at the beginning of the
observational period for each summer season, while the first 24 hours are discarded as a spin-up period. We chose 60 vertical
levels with a model top level at 50 hPa. We use a time step of 2.2 s for the most inner domain ($d_4$) and save the selected set of
variables as hourly and daily averages.

All WRF model runs use the same forcing (ERA5) and input data on land characteristics, such as topography and land
categories (Tables 2 and 3). Examples of the land-cover data used for WRF runs at Conrad and Kaskawulsh glaciers are shown
in Figure 4, while examples for the topography data are shown in Figure S1. Land cover data are taken from the European Space
Agency (ESA) Climate Change Initiative (CCI) dataset (ESA, 2017), but are converted to the 24 United States Geological
Survey (USGS) land use categories (Anderson et al., 1976) implemented in WRF. Initially, the land category for a few grid
cells overlapping with our glacier locations in $d_3$ and $d_4$ domains did not correctly display snow/ice category, but showed
bare ground tundra or evergreen needle-leaf forest category instead. These incorrect categories were manually corrected to the
snow/ice land category. The elevation of the AWSs in reality differs from the elevation of grid cells representing these AWS
locations in ERA5, ERA5-Land and WRF (Table S1), with the smallest differences, as expected, for the $d_4$ domain (1.1 km grid

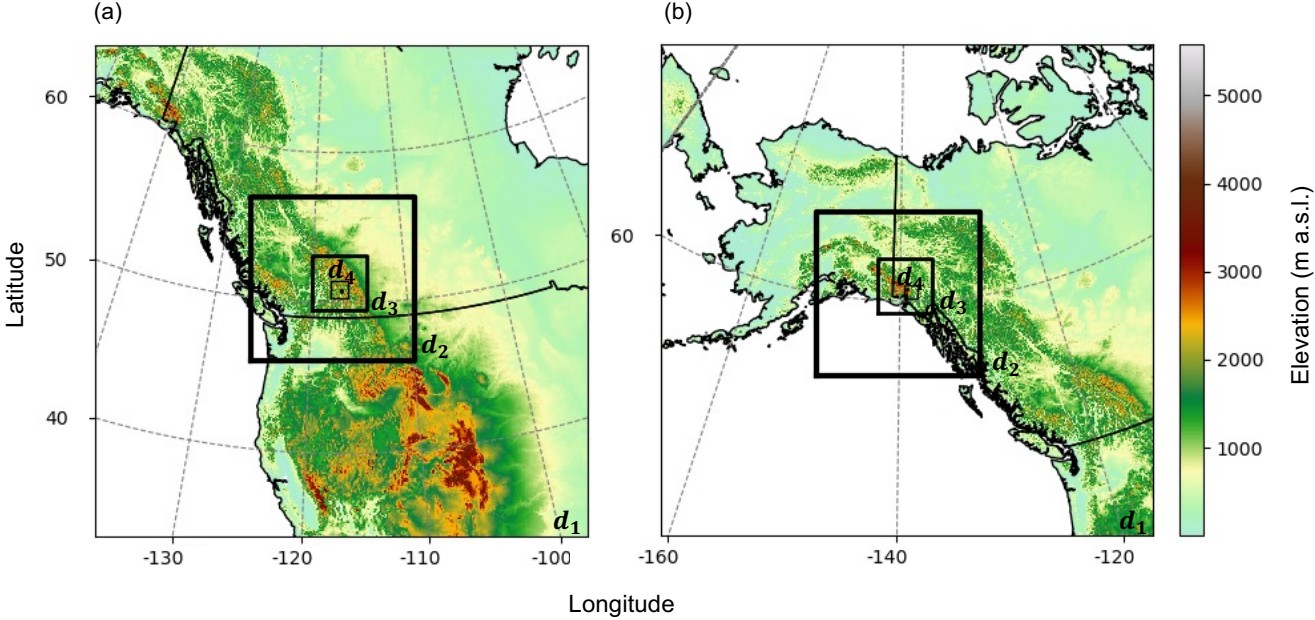

**Figure 3.** Topography map of the region with the borders of the nested domains used in the WRF model setup (a) Conrad, and (b) Kaskawulsh glacier: $d_1$ (the outer most domain) with 30 km grid spacing, $d_2$ with 10 km grid spacing, $d_3$ with 3.3 km grid spacing, and $d_4$ (the inner most domain) with 1.1 km grid spacing including the glacier (black dot). The topography maps were created from USGS GMTED2010 DEM (EROS, 2017).

spacing) in WRF. As our sites are located in complex mountainous terrain, slope and shadow effects on shortwave radiation are activated in WRF. The sea-surface temperature is updated daily in WRF, with hourly input data from ERA5 reanalysis. The WRF model was run on our department's high-performance computing cluster using three nodes (each with 20 cores) per glacier site.

The WRF model comes with various options for physics parameterizations (Skamarock et al., 2019), but previous glacier studies with WRF have used some parameterization schemes more often than others. For example, the most commonly used schemes in glaicer studies include RRTMG (Iacono et al., 2008), CAM (Collins et al., 2004), Dhudia (Dudhia, 1989) and Goddard (Max and Suarez, 1994; Matsui et al., 2018) for radiation, the Grell 3D Ensemble (Grell, 1993; Grell and Dévényi, 2002), the Kain-Fritsch (Kain, 2004) and the Betts-Miller-Janjić (Janjić, 1994) schemes for cumulus convection, and the Mor-
rison two-moment (Morrison et al., 2009), the Thompson (Thompson et al., 2008) and the updated aerosol–aware Thompson-Eidhammer (Thompson and Eidhammer, 2014) schemes for microphysics. The local-closure Mellor–Yamada–Nakanishi–Niino (MYNN) level 2.5 (Nakanishi and Niino, 2006, 2009; Olson et al., 2019) and Mellor-Yamada-Janjic (Janjić, 1994; Mesinger, 1993) schemes, as well as the non-local closure Yonsei University (Hong et al., 2006) scheme have been most commonly used for boundary layer, and the revised MM5 (Jiménez et al., 2012) and Eta Similarity (Monin and Obukhov, 1954; Janjić,
1994, 1996, 2002) schemes for surface layer. The Unified Noah (Tewari et al., 2004) and Noah-MP (Niu et al., 2011; Yang

**Table 2.** WRF model setup for this study. The WRF physics parameterization schemes are given in Table 3.

| Model configuration | | Lateral boundaries and input | |
|---|---|---|---|
| Simulation period | Conrad: 19 Jun–24 Aug 2016 | Forcing data | ERA5[1] (30 km) |
| | 17 Jul–05 Sep 2015 | Land cover | ESA CCI[2] (300 m) |
| | Nordic: 12 Jul–27 Aug 2014 | Topography | USGS GMTED2010[3] (1 km) |
| | Castle: 22 Aug–15 Sep 2012 | **Dynamics** | |
| | Kaskawulsh: 30 Jun–26 Aug 2019 | | |
| Time step | 60 s, 20 s, 6.7 s, 2.2 s | Vertical velocity damping | On |
| Spin-up time | 24 hours | Horizontal diffusion | Computed in physical space |
| Map projection | Lambert Conformal Conic | 6th-order numerical diffusion | On, with prohibited up-gradient diffusion |
| Horizontal grid spacing | 30 km: 121 × 121 grid points, | | |
| | 10 km: 121 × 121, | Damping coefficient | 0.02 |
| | 3.3 km: 121 × 121, | **Model physics** | |
| | 1.1 km: 121 × 121 (Nordic, | Sea-surface temperature update | Hourly from ERA5 |
| | Kaskawulsh: 91 × 91) | | |
| Vertical levels | 60 eta-levels | Effects on shortwave radiation | Slope effects and neighboring-point shadow effects |
| Model top | 50 hPa | | |
| | | Sea-ice albedo | Function of air and skin temperature and snow[9] |

[1] Hersbach et al. (2018)
[2] ESA (2017)
[3] EROS (2017)
[9] Mills (2011)

et al., 2011) land surface models are most commonly used in glacier studies. Noah-MP, which is a more sophisticated version of Unified Noah, includes multiple snow layers, representing percolation, retention, and refreezing of meltwater within the snowpack rather than in the snow–atmosphere and snow–soil interface as is the case with Unified Noah (Suzuki and Zupanski, 2018). The WRF simulations over non-glacierized terrain are shown to vary substantially depending on which of the two land
surface models is used (Milovac et al., 2016).

We chose our initial set of parameterizations based on those most commonly used in previous glacier studies (e.g., Mölg and Kaser, 2011; Claremar et al., 2012; Mölg et al., 2012; Collier et al., 2013, 2015). This set of parameterizations represents our reference (REF) model configuration (Table 3), while we also test different parameterizations as part of our sensitivity analysis. In this analysis, we perform 25 independent WRF runs, each with only one different parameterization scheme from those used
in REF (Table S2). The different parameterizations are selected from a set of previously used ones in a range of WRF glacier studies as well as a study that used WRF for hydrological modeling across Western Canada (Erler et al., 2014). For each set

**Table 3.** WRF physics parameterizations, for different physical processes, used in the three configurations: REF, minNRMSE and TOPSIS. In the parameterization for the cumulus process, the on/off label in brackets refers to the parameterization being switched 'on' or 'off' in each of the WRF domains $d_1$ (30 km) – $d_2$ (10 km) – $d_3$ (3.3 km) – $d_4$ (1.1 km).

| Process | REF | minNRMSE | TOPSIS |
|---|---|---|---|
| **Microphysics** | Thompson[1] | Thompson | Thompson |
| **Land Surface Model** | Noah-MP[2] | Unified Noah[3] | Noah-MP |
| **Longwave Radiation** | RRTMG[5] | RRTM[4] | RRTM |
| **Shortwave Radiation** | RRTMG | Dhudia[6] | Dhudia |
| **Cumulus** | Grell 3D Ensemble[7] (on – on – off – off) | Grell 3D Ensemble (on – on – on – off) | Betts–Miller–Janjic[8] (on – on – on – on) |
| **Planetary Boundary Layer** | MYNN [9] Level 3 | MYNN Level 3 | MYNN Level 3 |
| **Surface Layer** | MYNN | MYNN | MYNN |

[1] Thompson et al. (2008)
[2] Niu et al. (2011); Yang et al. (2011)
[3] Tewari et al. (2004)
[4] Mlawer et al. (1997)
[5] Iacono et al. (2008)
[6] Dudhia (1989)
[7] Grell (1993); Grell and Dévényi (2002)
[8] Janjić (1994)
[9] Nakanishi and Niino (2006, 2009); Olson et al. (2019)

of parameterizations, WRF is run for a period of seven days (six days after discarding 24 hours of spin-up time) at the four sites: Conrad glacier in 2015 and in 2016 (accumulation and ablation zone), and Nordic glacier in 2014. The six-day periods are selected randomly to represent different time windows throughout the early, middle, and late melt season.

We use these sensitivity runs to investigate whether there is a better configuration than REF, i.e., a best-performing configuration for our study sites. To this end, we evaluate the output from each sensitivity run against our AWS observations over the same six-day periods. The evaluation is performed for the meteorological variables relevant for the SEB modeling (described in the next section). Our goal is to determine the best-performing sensitivity run in each category of physics parameterization schemes (radiation, cumulus convection, microphysics, planetary boundary, surface layer, and land surface model). We focus
on the following two approaches:

      **minNRMSE configuration:** We determine the best-performing sensitivity run in each category of physics parameterizations as the one with minimum Normalized Root Mean Square Error (NRMSE) in the modeled melt energy, where the reference data is the timeseries of modeled daily melt energy derived from the AWS data. The final optimal configuration, labeled as minNRMSE, includes the best parameterization schemes from each of the categories.

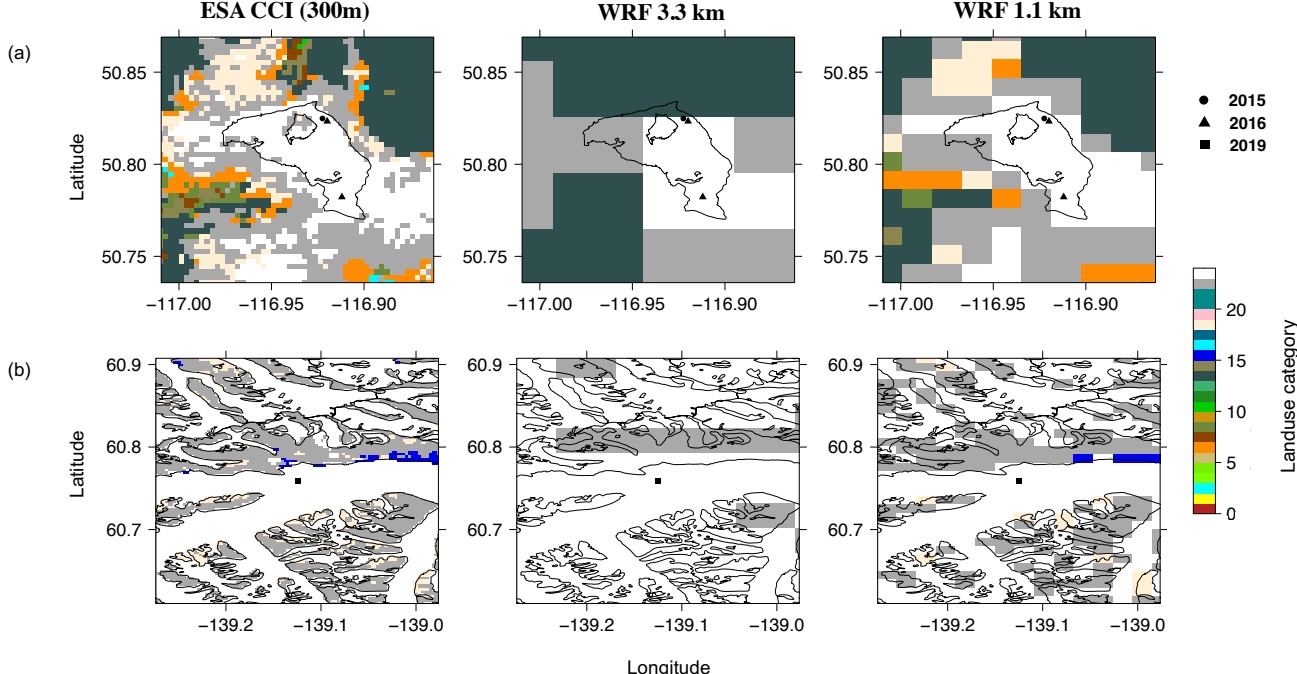

**Figure 4.** Land cover categories for the domain covering (a) Conrad and (b) Kaskawulsh glacier from ESA CCI Land Cover at 300 m grid spacing (ESA, 2017) in comparison to the land categories from WRF at 3.3 km and 1.1 km grid spacing. Markers indicate the AWS sites in different years. The outlines of the glaciers (black lines) are taken from the Randolph Glacier Inventory (RGI V6; RGI Consortium, 2017). There are 24 land-cover categories, while the category enumerated as 24 corresponds to ice/snow (colored as white in the figures). On the map with Kaskawulsh glacier the neighboring glaciers are also shown.

**TOPSIS configuration:** We determine the best-performing sensitivity run in each category using the multi-criteria decision-making method known as the Technique for Order Preference by Similarity to the Ideal Solution (TOPSIS), originally introduced in Hwang and Yoon (1981). TOPSIS aims to identify the best alternative based on the shortest geometric distance from the positive ideal solution and the longest geometric distance from the negative ideal solution. Instead of using just one evaluation metric, such as NRMSE, we introduce additional metrics: Spearman rank correlation coefficient ($r_{sp}$), Normalized Nash–Sutcliffe model efficiency coefficient (NNSE; Nash and Sutcliffe, 1970; Nossent and Bauwens, 2012), and Mean Absolute Percentage Error (MAPE). The evaluation metrics are all weighted equally across the sites. Similarly to the minNRMSE method above, the reference data is the modeled melt energy assessed from AWS observations. This method has been applied in multiple studies to find the best physics parameter schemes in WRF (e.g., Stergiou et al., 2017; Wang et al., 2021). We follow the TOPSIS methodology described in Tzeng and Huang (2011).

## 2.4 Surface energy balance model

Surface melt modeling through SEB accounts for the surface energy budget at the glacier-atmosphere interface, and thus calculates the energy available for melt once the surface is at the melting point. Here we use a relatively simple SEB model that considers only the key contributors to total surface melt energy over a summer season at mid-latitude glaciers (Hock, 2005). The modeled energy available for melt, $Q_M$ (Wm$^{-2}$) at a given point on a glacier is derived as:

$$Q_M = K_{in}(1 - \alpha) + L_{in} - \sigma T_0^4 + Q_H + Q_L, \tag{1}$$

where $K_{in}$ and $L_{in}$ are the incoming shortwave and longwave radiation, respectively, $\alpha$ is the surface albedo, and $Q_H$ and $Q_L$ are the sensible and latent heat fluxes, respectively. $T_0$ is the glacier's surface temperature and $\sigma$ is the Stefan-Boltzmann constant. The outgoing longwave radiation is approximated using the Stefan-Boltzmann law with emissivity set to unity, where $T_0$ is set to $0\,°C$ as it is also confirmed with measurements at our glacier sites (Fitzpatrick et al., 2017, 2019; Lord-May and Radić, 2023). All variables in the model are represented as their daily mean values. Fluxes are defined as positive (negative) when directed towards (outwards) the surface. Once the surface temperature reaches $0\,°C$ and stays at the melting point throughout the summer season, a positive $Q_M$ in Eq. 1 drives melt.

Given our focus on the key seasonal SEB components, we neglect the ground heat flux and the heat flux from rain, since both have been shown to give negligible contributions to the total seasonal melt at mid-latitude glaciers (Sicart et al., 2005; Andreassen et al., 2008; Gillett and Cullen, 2011), as well as at our study sites (Fitzpatrick et al., 2017; Fitzpatrick, 2018; Lord-May and Radić, 2023). The rain heat flux, however, can be a substantial contributor (up to $20\,\%$) to daily melt energy on a day with extreme rainfall (Fitzpatrick et al., 2017), but the uncertainty in the model used to assess the rain heat flux is relatively large (Hock, 2005; Fitzpatrick et al., 2017). For these reasons, we neglect the heat flux in the SEB model, but will include precipitation, both as rainfall and snowfall, in the evaluation analysis.

For the simplicity of the model we also neglect empirical correction schemes commonly applied to the shortwave radiation fluxes, such as separation into direct and diffuse components as in Hock and Holmgren (2005). These corrections, however, are shown to have minor effect on simulated seasonal melt at our sites in the interior of British Columbia (Fitzpatrick et al., 2017). The turbulent heat fluxes are calculated using the bulk aerodynamic method:

$$Q_H = \frac{p}{p_0} \, \rho_a \, c_p \, C_H \, U_z \, (T_z - T_0), \tag{2}$$

$$Q_L = \frac{0.622}{p_0} \, \rho_a \, L_v \, C_L \, U_z \, (e_z - e_0), \tag{3}$$

where $U_z$ is the mean near-surface wind speed at height $z$, and $T_z$ and $e_z$ are mean air temperature and vapor pressure at height $z$ (2 m at our AWSs), respectively. $e_0$ is the mean vapor pressure, $\rho_a$ is air density, $c_p$ is specific heat capacity of air at constant pressure and $L_v$ is the latent heat of vaporization of snow or ice. $p_0$ is the air pressure at standard sea level, $p$ is the actual air pressure, and $C_H$ and $C_L$ are dimensionless exchange coefficients for sensible and latent heat, respectively. The exchange coefficients are parameterized following the Monin-Obukhov stability theory (Monin and Obukhov, 1954) and depend on the surface roughness for momentum ($z_{0v}$), temperature ($z_{0T}$) and humidity ($z_{0q}$), and on the atmospheric stability conditions in

the surface boundary layer. We use constant values for these three roughness lengths for each site, which have been adopted from the EC-derived values determined from previous studies at these sites (Table S3; Radić et al., 2017; Fitzpatrick et al., 2017, 2019; Lord-May and Radić, 2023), while for the stability corrections, based on the assessed stability conditions, we use the functions applied previously in Fitzpatrick et al. (2017). The order of magnitude in these EC-derived values for roughness lengths ($z_{0v} = 10^{-3}$ m, $z_{0T} = z_{0q} = 10^{-5}$ m) agree with commonly assumed values for glaciers in mid-latitudes (Hock, 2005). Vapor pressure $e_z$ at height $z$ is calculated from the relative humidity $RH$ at height $z$ using the August-Roche-Magnus formula (Alduchov and Eskridge, 1997).

## 2.5 Evaluation analysis

The primary goal of the evaluation analysis is to assess the performance of the SEB model, forced with either ERA5 or WRF data, in simulating seasonal melt energy at our sites. To do so, we evaluate the total simulated energy available for melt ($Q_M$; Eq. 1), as well as the daily timeseries of $Q_M$, as calculated from the SEB model forced with the reanalyses (ERA5, ERA5-Land), as well as with the WRF output, against the reference calculations when the same SEB model is forced with the AWS data. Thus, the input for the SEB model, i.e. the atmospheric variables $K_{in}$, $L_{in}$, $T$, $RH$, and $U$, are taken from: (1) the AWS at each site, representing the reference or true values, (2) ERA5, (3) ERA5-Land, and (4) WRF at grid spacings of 3.3 and 1.1 km, using each of the three configurations (REF, minNRMSE, and TOPSIS). For the reanalysis and WRF, only the data from the grid cell covering each study site is used.

As we are interested in the evaluation of meteorological rather than surface variables (albedo and surface roughness), we use in-situ observations of daily surface albedo and seasonally-averaged roughness lengths in the SEB model. These surface variables could have been taken directly from the reanalysis and WRF; however, we found that these values can differ substantially from the observed ones throughout the observational periods (Figure 5 and Table 4). The discrepancy between WRF and the observed albedo on glaciers, especially in the ablation zone, has also been noted in previous glacier studies (Collier et al., 2013; Eidhammer et al., 2021). Thus, to avoid any evaluation biases originating in poorly assigned surface variables, we stick to the choice of using observed surface variables in the SEB model. We note that while we incorporate observed albedo into the SEB model, the inaccurately simulated albedo in WRF still influences the near-surface meteorological forcing fields. The observed daily surface albedo is calculated as the ratio of measured daily totals (in local daylight hours) of reflected and incoming shortwave radiation at each site. The incoming shortwave radiation at the surface is taken from these datasets without any further modifications (e.g., separation into direct and diffuse radiation).

Since the elevations of the grid cells from the reanalyses and WRF differ from the actual AWS elevations (Table S1), we perform a lapse-rate correction on the temperature data from these datasets. Here we do so by calculating a daily-averaged lapse rate from a regression between ERA5 temperature and geopotential height at multiple pressure levels for each glacier site. This timeseries of daily lapse rates is then used to correct the timeseries of daily 2 m air temperature ($T$) from the reanalyses and WRF over the observational period.

Near-surface wind speeds in the reanalyses and WRF are given at a height of 10 m above the surface, while the AWS wind data was measured at a height of 2 m above the surface. A common correction for the difference in wind speed heights is based

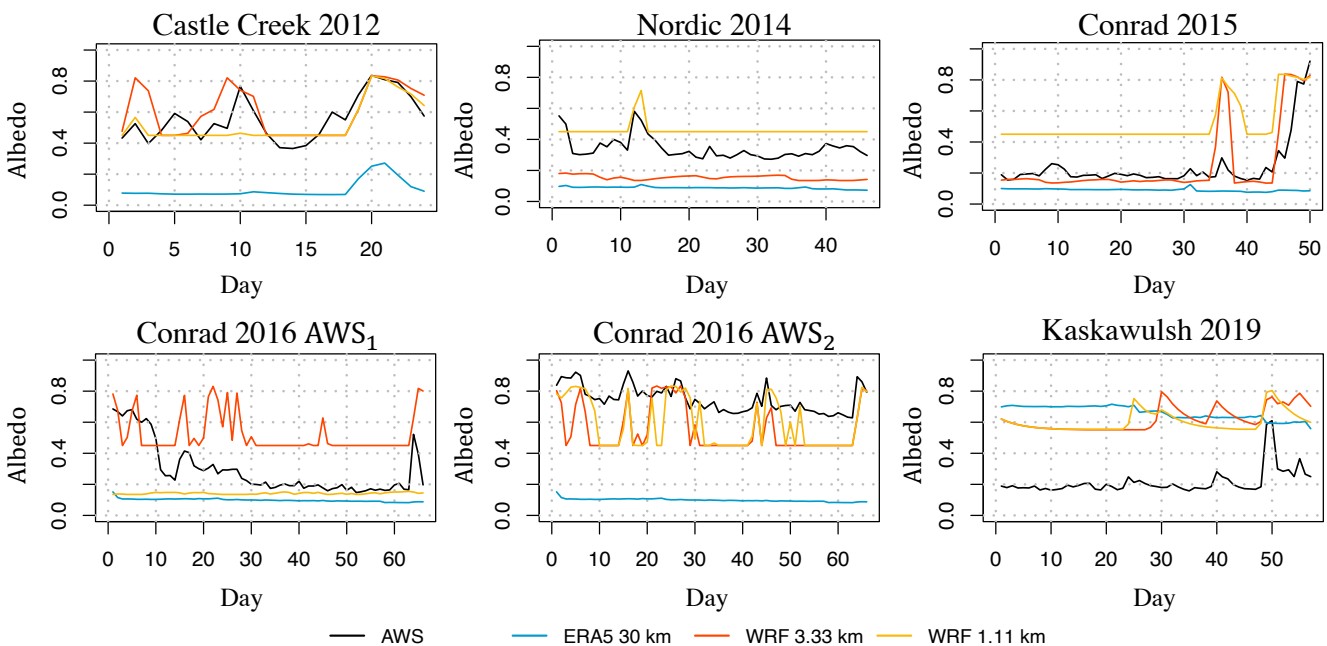

**Figure 5.** Modeled (ERA5, WRF at 3.3 km and WRF at 1.1 km) and observed (AWS data) timeseries of daily albedo over the observational period (starting at Day 1) at each site. WRF is run with the REF configuration.

**Table 4.** Mean seasonal roughness lengths [m] for momentum ($z_{0v}$) derived from the observations (AWS), ERA5 and WRF (3.3 km and 1.1 km) at each study site.

| Site | AWS | ERA5 (30 km) | WRF (3.3 km) | WRF (1.1 km) |
|---|---|---|---|---|
| Castle Creek 2012 | 0.003 | 0.067 | 0.002 | 0.002 |
| Nordic 2014 | 0.003 | 0.936 | 0.002 | 0.002 |
| Conrad 2015 | 0.003 | 1.169 | 0.002 | 0.002 |
| Conrad 2016 AWS$_1$ | 0.001 | 1.166 | 0.002 | 0.002 |
| Conrad 2016 AWS$_2$ | 0.003 | 1.166 | 0.002 | 0.002 |
| Kaskawulsh 2019 | 0.001 | 0.001 | 0.002 | 0.002 |

on the assumption of a logarithmic wind profile (e.g., Claremar et al., 2012; Giese et al., 2022), which rarely takes place at our study sites, especially those in the interior of British Columbia, where katabatic flow with low (<3 m above surface) wind speed maxima prevails during a summer season (e.g., Fitzpatrick et al., 2017; Radić et al., 2017). The correction based on the logarithmic wind profile may therefore introduce an additional bias (underestimation) of wind speed relative to the observed

wind speed at 2 m. We therefore chose not to correct for the height difference in the wind datasets. The remaining variables, $L_{in}$ and $RH$, are also taken directly from the reanalysis and WRF without any modifications.

For each glacier site, we evaluate how closely ERA5, ERA5-Land and WRF (1.1 km and 3.3 km) resemble the observed components of the SEB as well as the total energy for melt ($Q_M$) derived from the SEB model forced by AWS data. To do so, we use the same evaluation metrics as in the TOPSIS method ($r_{sp}$, NRMSE, MAPE, NNSE), and also add Normalized Mean

Bias Error (NMBE). While we evaluate the reanalysis and WRF performance in simulating day-to-day variability of those variables, our main focus is on evaluating their daily values as a mean over the whole observational period at each site.

In addition to the aforementioned variables, we also look into how well the reanalyses and WRF simulate the timeseries of daily precipitation ($P$), both in liquid and solid form. While an extreme rainfall event can present a strong contribution to daily melt energy (Fitzpatrick et al., 2017), fresh snowfall events over a summer season can substantially alter glacier

albedo and consequently the net radiative fluxes and the SEB (e.g., Hock, 2005; MacDougall et al., 2011; Marshall and Miller, 2020). Despite our choice to use the observed albedo in the SEB model, we look into how well the reanalysis and WRF capture the frequency of fresh snowfall events over the observational periods at our sites. To differentiate between rainfall and snowfall, we use a simple model that relies on the temperature threshold of 0 °C: rainfall (snowfall) is assumed when the near-surface temperature at the given site is above (below) the threshold. We note that the overall quality of in-situ precipitation

measurements, based on tipping bucket rain gauges, is likely to be lower relative to the quality of other measurements at our sites (Fitzpatrick et al., 2017). The rain gauges can have extensive underestimation of rainfall amounts (of up to 50 %), primarily due to acceleration of airflow over the top of the gauge, with other error sources including splashing, and the finite time required for the buckets to reset between tips during heavy rain (e.g., Devine and Mekis, 2008; Duchon and Biddle, 2010).

## 3 Results

### 3.1 Evaluation of meteorological variables

Here we evaluate the performance of ERA5, ERA5-Land and WRF (3.3 km and 1.1 km) in simulating the selected variables ($T$, $RH$, $P$, $U$, $K_{in}$, $L_{in}$) from the six study sites over their observational periods. A summary of results based on the relative error and NRMSE is shown in Table 5, while the results based on other evaluation metrics are shown in the Supplementary Information (Tables S4 and S5). Looking first at the results for the reanalysis data, we find that ERA5 and ERA5-Land yield

similar performance (difference of a few percent in NRMSE), with an overall slightly better performance, though not statistically significant (Table S5), of ERA5 over ERA5-Land. ERA5, and similarly ERA5-Land, are found to simulate the mean daily radiative fluxes ($K_{in}$ and $L_{in}$) relatively close to observations, with a relative error (overestimation) of 11 % in $K_{in}$ and no relative error in $L_{in}$ (Figure 6). The mean daily sensible heat flux from ERA5 is substantially underestimated (relative error of 87 %), mainly due to the substantial underestimation of mean wind speed (relative error of 64 %) despite the well simulated

lapse-rate corrected mean daily near-surface air temperature (overestimated by 14 %). Similarly, the mean daily latent heat flux is also substantially overestimated. However, as the contribution of the latent heat flux to the total melt energy is small (<5 %), the large errors here reflect the differences in small numbers (e.g., 0.5 W m$^{-2}$ versus 3 W m$^{-2}$). Correlating the daily timeseries

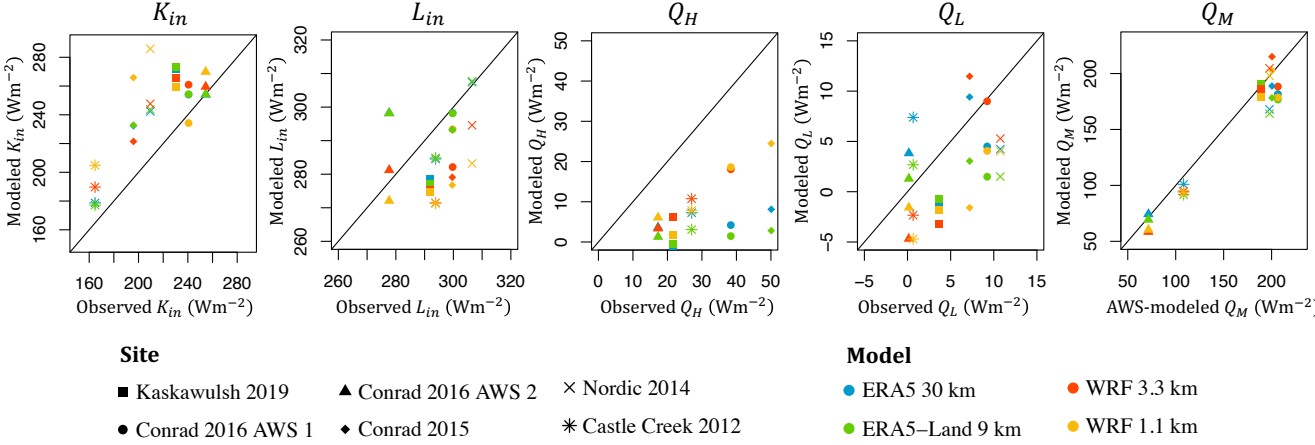

**Figure 6.** Modeled (ERA5, ERA5-Land, WRF at 3.3 km and WRF at 1.1 km) versus observed (AWS data) daily averages over the observational period of incoming shortwave ($K_{in}$) and longwave ($L_{in}$) radiation, sensible ($Q_H$), latent ($Q_L$) heat fluxes and melt energy ($Q_M$). The melt energy is estimated according to the SEB model (Eq. 1). WRF is run with the REF configuration.

between ERA5 or ERA5-Land and AWS equivalent variables reveals that all correlations are statistically significant (p-value < 0.05) except for $Q_L$ for ERA5, and except for $U$, $Q_H$ and $Q_L$ for ERA5-Land (Table S4). Seasonally-averaged total daily

precipitation ($P$) from ERA5 is overestimated at some glacier sites (relative error of 130.3 % for Kaskawulsh 2019 and 84.5 % for Conrad 2016 AWS$_1$), while underestimated at other sites (relative error of −37.9 % for Conrad 2016 AWS$_2$, −33.2 % for Conrad 2015, and −18.7 % for Nordic 2014; Figures S2 and S3). On the other hand, the modeled timeseries of daily precipitation all show statistically significant correlation (p-value < 0.05) with the observed timeseries (Figure S2). The frequency of days with heavy snowfall is overestimated in the ablation zones (Castle Creek 2012, Kaskawulsh 2019) and underestimated

in the accumulation zone (Conrad 2016 AWS$_2$). On average, ERA5-Land precipitation simulations perform worse than ERA5 (mean overestimation of 35.6 % in ERA5-Land versus 20.3 % in ERA5 across all sites).

The results from WRF reveal similar performance patterns as those for the reanalysis, but with some differences depending on the three configurations (REF, minNRMSE and TOPSIS; Table 5). For the REF configuration, the mean incoming longwave radiation is slightly underestimated (5 % for WRF at 3.3 km and 1.1 km), while the mean sensible heat flux is substantially un-

derestimated (43 % for WRF at 3.3 km and 64 % for WRF at 1.1 km; Figure 6). The REF configuration yields an overestimation in mean $K_{in}$ (12 % for WRF at 3.3 km and 19 % for WRF at 1.1 km), while the minNRMSE configuration gives an underestimation (16 % for WRF at 3.3 km and 11 % for WRF at 1.1 km), and TOPSIS only a small relative error (4 % underestimation for WRF at 3.3 km and 1 % overestimation for WRF at 1.1 km). Correlating the daily timeseries between equivalent variables from WRF and AWS reveals that all correlations are statistically significant (p-value < 0.05) except for $U$ (Table S5). In con-

trast to ERA5 and ERA5-Land, WRF does yield statistically significant correlations (p-value < 0.05) for both $Q_H$ and $Q_L$. WRF at 1.1 km underestimates the seasonally-averaged total precipitation ($P$) for all glacier sites (relative errors ranging from −3.1 % for Conrad 2015 to −45.0 % for Kaskawulsh 2019 with the REF configuration) except for Conrad 2016 AWS$_1$, where

**Table 5.** Relative difference (%) between modeled and observed seasonally-averaged values, as well as NRMSE between modeled and observed daily values of: air temperature ($T$), relative humidity ($RH$), total precipitation ($P$), wind speed ($U$), incoming shortwave ($K_{in}$) and longwave ($L_{in}$) radiation, sensible ($Q_H$) and latent ($Q_L$) heat fluxes, and total melt energy ($Q_M$). The melt energy is estimated according to the SEB model (Eq. 1). The WRF runs are based on the three configurations of physics parameterizations: REF, minNRMSE and TOPSIS. For comparison, we also include the results of the ensemble-mean, where $T$, $RH$, $P$, $U$, $K_{in}$ and $L_{in}$ are derived as a mean across the three configurations. The turbulent heat fluxes and $Q_M$ in the ensemble-mean are derived according to the aerodynamic bulk method (Eqs. 2 and 3) and SEB model (Eq. 1), respectively. The results of each evaluation metric are shown as the mean ($\pm$ one standard deviation) across the six study sites, with equal weighing of each site. For seasonally-averaged values of $Q_M$, we only take into account positive values of $Q_M$ that drive melt. Values in bold highlight the best performing model for the given variable according to the metric used.

| Variable | ERA5 30 km | ERA5-Land 9 km | WRF 3.3 km | | | | WRF 1.1 km | | | |
|---|---|---|---|---|---|---|---|---|---|---|
| | | | REF | minNRMSE | TOPSIS | Ensemble | REF | minNRMSE | TOPSIS | Ensemble |
| **Relative Error (%)** | | | | | | | | | | |
| $T$ | 14 ± 24 | 23 ± 23 | **3** ± 37 | -21 ± 48 | 9 ± 37 | **-3** ± 40 | 6 ± 30 | -25 ± 39 | 11 ± 29 | **-3** ± 32 |
| $RH$ | 30 ± 12 | 28 ± 12 | **-2** ± 14 | 17 ± 22 | -6 ± 12 | 3 ± 14 | -9 ± 11 | 10 ± 16 | -12 ± 9 | -4 ± 12 |
| $P$ | 20 ± 70 | 36 ± 99 | 29 ± 88 | 46 ± 101 | **1** ± 45 | 25 ± 77 | -5 ± 51 | 12 ± 53 | -15 ± 27 | -3 ± 43 |
| $U$ | -64 ± 10 | -74 ± 6 | -26 ± 18 | -42 ± 12 | **-22** ± 20 | -30 ± 15 | -44 ± 16 | -46 ± 14 | -42 ± 15 | -44 ± 14 |
| $K_{in}$ | 11 ± 8 | 11 ± 8 | 12 ± 6 | -16 ± 16 | -4 ± 9 | -3 ± 9 | 19 ± 16 | -11 ± 18 | **1** ± 11 | 3 ± 14 |
| $L_{in}$ | **0** ± 4 | -1 ± 4 | -5 ± 3 | **0** ± 4 | -3 ± 3 | -3 ± 3 | -5 ± 3 | -1 ± 3 | -3 ± 2 | -3 ± 2 |
| $Q_H$ | -87 ± 11 | -95 ± 5 | -43 ± 38 | -81 ± 14 | **-40** ± 40 | -58 ± 22 | -64 ± 16 | -82 ± 13 | -64 ± 16 | -73 ± 13 |
| $Q_L$ | 520 ± 977 | **113** ± 335 | -613 ± 1203 | -115 ± 207 | -804 ± 1533 | -399 ± 758 | -383 ± 453 | -235 ± 298 | -531 ± 702 | -368 ± 437 |
| $Q_M$ | -6 ± 7 | -10 ± 7 | **-5** ± 10 | -28 ± 10 | -15 ± 15 | -26 ± 16 | -8 ± 7 | -27 ± 10 | -19 ± 8 | -29 ± 15 |
| **NRMSE (%)** | | | | | | | | | | |
| $T$ | 22 ± 10 | 24 ± 10 | 23 ± 8 | 28 ± 6 | 24 ± 8 | **21** ± 8 | **21** ± 13 | 25 ± 6 | 22 ± 14 | **21** ± 10 |
| $RH$ | 50 ± 8 | 47 ± 7 | 32 ± 22 | 47 ± 20 | 32 ± 23 | 32 ± 20 | 33 ± 22 | 35 ± 13 | 35 ± 21 | **29** ± 17 |
| $P$ | **21** ± 5 | 24 ± 9 | 31 ± 14 | 35 ± 16 | 30 ± 13 | 28 ± 12 | 27 ± 10 | 30 ± 13 | 27 ± 12 | 24 ± 10 |
| $U$ | 55 ± 15 | 62 ± 16 | 40 ± 8 | 44 ± 9 | **38** ± 8 | **38** ± 9 | 43 ± 10 | 45 ± 13 | 44 ± 9 | 43 ± 10 |
| $K_{in}$ | **17** ± 2 | **17** ± 3 | 26 ± 2 | 30 ± 11 | 26 ± 5 | 23 ± 4 | 29 ± 5 | 29 ± 9 | 27 ± 4 | 24 ± 2 |
| $L_{in}$ | **18** ± 6 | **18** ± 6 | 30 ± 7 | 28 ± 5 | 32 ± 8 | 27 ± 6 | 31 ± 8 | 27 ± 7 | 30 ± 8 | 26 ± 8 |
| $Q_H$ | 39 ± 13 | 41 ± 12 | 31 ± 8 | 35 ± 8 | **30** ± 7 | 31 ± 9 | 31 ± 9 | 35 ± 9 | 32 ± 9 | 34 ± 10 |
| $Q_L$ | 25 ± 6 | 23 ± 8 | 23 ± 5 | 22 ± 6 | 24 ± 3 | 22 ± 6 | 21 ± 5 | 21 ± 6 | 22 ± 5 | **20** ± 6 |
| $Q_M$ | **14** ± 4 | 15 ± 4 | 18 ± 3 | 23 ± 7 | 21 ± 4 | 19 ± 4 | 16 ± 3 | 22 ± 6 | 19 ± 3 | 18 ± 3 |

$P$ is overestimated by 94.0 %. The modeled timeseries of daily precipitation shows statistically significant correlation (p-value < 0.05) with the observed timeseries for all sites except for Kaskawulsh glacier (Figure S2). WRF tends to overestimate the frequency of days with heavy snowfall in both the ablation zone (Castle Creek 2012, Kaskawulsh 2019) and the accumulation zone (Conrad 2016 AWS$_2$; Figure S3). Additionally, it also overestimates the frequency of days with light rainfall (< 2.5 mm). On average, precipitation values from WRF at 3.3 km perform significantly worse than from WRF at 1.1 km (29.0 % versus −5.0 % relative error with the REF configuration; Table 5).

When comparing the performance of the three WRF configurations at 1.1 km for the whole observational periods at each study site, we find that no consistent pattern of outperformance or underperformance is found across the variables and glacier

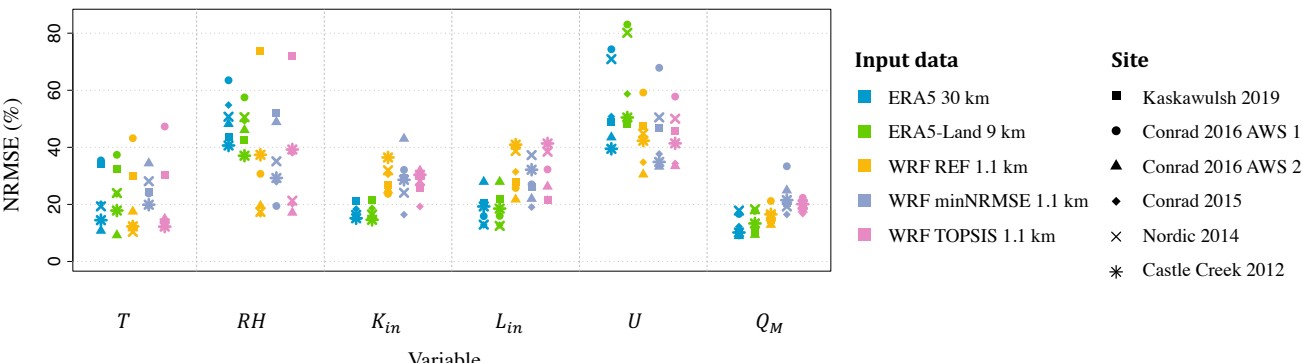

**Figure 7.** Performance of the reanalysis (ERA5 and ERA5-Land), and WRF at 1.1 km grid spacing, according to NRMSE calculated for each study site for the following variables: air temperature ($T$), relative humidity ($RH$), incoming shortwave ($K_{in}$) and longwave ($L_{in}$) radiation, wind speed ($U$) and total melt energy ($Q_M$). The melt energy is estimated according to the SEB model (Eq. 1). The WRF runs are presented for REF, minNRMSE and TOPSIS configurations.

sites (Figure 7). Overall, there is a smaller spread in NRMSE across the sites for the minNRMSE configuration relative to the other two configurations, but even this finding does not hold for each variable (Table 5). Notably, the simulation of $RH$ at Kaskawulsh glacier is particularly poor (NRMSE of 70 %) for both TOPSIS and REF, while for the other sites the NRMSE in $RH$ does not exceed 50 %. Wind speed is another variable with large NRMSE (>30 % across all the sites), most pronounced

in the ablation area of Conrad glacier in 2016 (NRMSE of 57 %, while at almost the same location on Conrad glacier in 2016 the NRMSE is much smaller with 38 %). Despite the poor simulation of $U$, all three WRF configurations yield similar and statistically significant correlations (p-value < 0.05) between modeled and AWS-modeled daily $Q_H$ (Table S5). The results highlight relatively large variability in the WRF model performance across the study sites and observational periods, as well as across the selected variables. For comparison, we also looked into the WRF performance as an ensemble-mean across the

three WRF configurations (Table 5). To this end, we averaged the results from the three configurations (REF, minRMSE, and TOPSIS) for each variable. Relative to at least two individual members, the ensemble-mean has slightly improved performance (smaller relative errors) for $T$, $RH$, $P$ and $K_{in}$. Wind speed and turbulent heat fluxes still remain substantially underestimated.

     Of the variables that play an important role in the SEB model, $RH$ and $U$ have, on average, the largest errors for the two reanalyses and WRF (Figure 7). The relatively poor simulation of wind speed and direction, in both reanalyses and WRF,

reveals the inability of the models to capture the katabatic (downglacier) flow that prevails during summer months at the glacier sites. Failing to resolve the strong downslope wind speeds with maxima close to the glacier surface, the reanalysis and WRF substantially underestimate the wind speed (Figure 8). While WRF does resolve the wind speed better than the reanalysis, the underestimation of wind speed in WRF is still substantial (44 % in WRF at 1.1 km and REF run, relative to 64 % in ERA5). During episodes of synoptic storms, when the katabatic flow is interrupted, there is a slightly better agreement between the

modeled and observed wind speed and direction across the glacier sites for both ERA5 and WRF (Figure 8). However, these

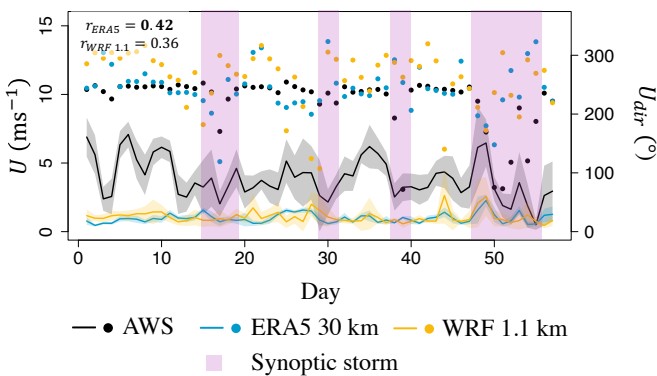

**Figure 8.** Modeled (ERA5 and WRF at 1.1 km) versus observed (AWS data) timeseries of daily-averaged wind speed ($U$, line), including the range of one standard deviation (shaded), and daily-averaged wind direction ($U_{dir}$, dots) for Kaskawulsh glacier in 2019. Time windows with observed synoptic storms are marked in purple vertical shading. Bold values of $r_{sp}$ indicate a statistically significant correlation at the 5 % confidence level. WRF is run with the REF configuration.

episodes are relatively rare and short-lasting to make any substantial difference in the overall model performance in simulating $U$.

We also investigated the use of surface $Q_H$ and $Q_L$ as outputted directly from the reanalysis and WRF into the SEB model, rather than calculating those fluxes with our bulk method. In WRF, these fluxes are derived through a local or non-local closure
scheme in the planetary boundary and surface layer, depending on the parameterizations used (Skamarock and Klemp, 2008). When $Q_H$ is directly taken from ERA5, the NRMSE of $Q_H$ is 83 %, which is twice as large as the original error when $Q_H$ is calculated with the bulk method. In WRF at 1.1 km, the error in $Q_H$ is increased from 31 % when the bulk method is used to 60 %, while the error for $Q_L$ is increased from 21 % to 54 % with the REF configuration. For Kaskawulsh glacier, the largest glacier among our study sites, the performance of simulated $Q_H$ and $Q_L$ directly from ERA5 is similar or only slightly worse
(few percent) than the performance based on the bulk method. Across all the sites, taking $Q_H$ and $Q_L$ directly from ERA5 leads to an increased underestimation of mean $Q_M$ from 6 % in the original estimate to 72 %. For WRF at 1.1 km, the relative error in $Q_M$ increased from 8 % in the original estimate to 17 %. These results justify our choice to assess the turbulent heat fluxes via the bulk method instead of taking them directly from the reanalyses and WRF.

## 3.2 Sensitivity analysis

Our sensitivity analysis to the choice of parameterizations in WRF consists of running the WRF model with the REF configuration over the selected six-day periods and study sites, and altering only one physics scheme in the REF confirmation per run. This process yields in total 25 WRF independent runs (Table S2), including the one with the REF configuration, whose output is then evaluated against the AWS observations over the same sites and time windows using NRMSE (Figures 9 and S4). The larger the range of NRMSE across these runs, the larger the sensitivity to the choice of the schemes.

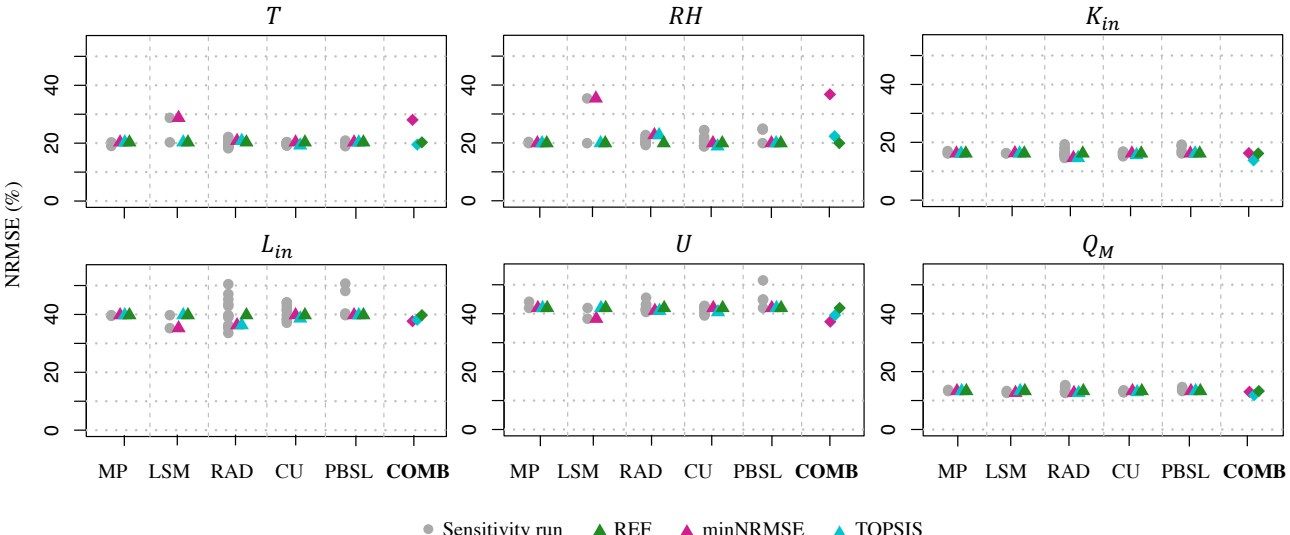

**Figure 9.** Performance of the evaluated 25 sensitivity runs, as well as the runs with the three configurations (REF, minNRMSE, TOPSIS), according to the NRMSE for the following variables: air temperature ($T$), relative humidity ($RH$), incoming shortwave ($K_{in}$) and longwave ($L_{in}$) radiation, wind speed ($U$) and total melt energy ($Q_M$). The parameterization schemes are split into the following categories: microphysics (MP), land surface model (LSM), radiation (RAD, including shortwave and longwave), cumulus (CU), and planetary boundary and surface layer (PBSL). Configuration details on these runs are given in Tables 3 and S2. In each category, the scheme that is being used in the three configurations (REF, minNRMSE, TOPSIS) is marked with a triangle, while the performance of the three configurations is presented in each plot under COMB and marked with a diamond.

Altering the land surface model between Unified Noah and Noah-MP yields the largest impact, i.e. the largest range of NRMSE, in the simulations of $RH$ (NRMSE in the range of 20–35 %) and $T$ (NRMSE in the range of 20–29 %), followed by the simulations of $U$ (NRMSE in the range of 38–42 %) and $L_{in}$ (NRMSE in the range of 35–40 %). Altering the radiation scheme, where 12 different schemes were tested, made the largest impact on the simulation of $L_{in}$ (NRMSE in the range of 34–50 %), while having relatively small impact (<4 % range) on the simulation of all other variables. Altering the cumulus

scheme, with three different cumulus schemes and three parameterization configurations each tested, made the largest impact on simulations of $L_{in}$ (NRMSE in the range of 37–44 %) and $RH$ (NRMSE in the range of 19–24 %), while having relatively small impact (<4 % range) on the simulation of the remaining variables. The sensitivity to the choice of the four schemes for the planetary boundary layer is the largest for simulating $L_{in}$ (NRMSE in the range of 40–51 %) and $U$ (NRMSE in the range of 42–52 %) with relatively small sensitivity in other variables. Finally, none of these altered WRF configurations yields a

strong impact on the calculated $Q_M$ from the SEB model (Eq. 1): NRMSE is in the range of 12–15 % (Figure 9). Note that when different optimal configurations were used (REF, minNRMSE, TOPSIS) the mean NRMSE for simulating $Q_M$ over all the sites and observational periods were in the range of 16–22 % for WRF at 1.1 km, and 18–23 % for WRF at 3.3 km (Table 5).

Looking at the performance of the 25 sensitivity runs, as well as the performance of the three "optimal" configurations, we identified the schemes for each category that consistently performed better in simulating the components of the SEB at our sites. For the microphysics, the best-performing is the Thompson scheme, while for the planetary boundary and surface layer the best-performing is the MYNN Level 3 scheme. For the cumulus scheme, the results were less conclusive in terms of identifying only one scheme that consistently performed better. Instead, we found that two cumulus schemes performed better than others: the Betts-Miller-Janjić scheme that is switched "on" in all domains, and the Grell 3D Ensemble scheme that is turned "off" only in the innermost domain ($d_4$), or the two inner domains ($d_3$ and $d_4$). For the radiation scheme, both the minNRMSE and TOPSIS method preferred the RRTM scheme for longwave radiation and the Dhudia scheme for shortwave radiation. The Noah-MP land surface model gave better results than Unified Noah, in particular for the simulations of near-surface temperature and relative humidity (Figure 9). In both land surface models, the glacier surface albedo is calculated as a weighted average of land ice albedo and snow albedo based on snow cover fraction (He et al., 2023). However, we modified the current Noah-MP albedo parameterizations for land ice, as the default options were too high for our glacier sites: the ice albedo values were changed from 0.80 to 0.6 for the visible spectrum, and from 0.55 to 0.3 for the near-infrared spectrum. No changes were applied to the albedo representations within Unified Noah.

In addition to the parameterization schemes, the WRF output is known to be sensitive to its 'nesting' configuration, including the choice of domain boundaries and their size and grid spacing within. Due to the nature of our study domain, we could not follow the general recommendation for placing each of the domain boundaries outside of complex terrain (Skamarock and Klemp, 2008). However, we assessed the sensitivity of WRF output to small latitudinal and longitudinal shifts (by few grid cells) of the domain boundaries for our two inner domains ($d_3$ and $d_4$) at Castle Creek, Conrad and Nordic glaciers. The results reveal a negligible difference in the WRF output at the study sites. Furthermore, changing the grid refinement ratio from the original 1:3 ($30\,\mathrm{km} - 10\,\mathrm{km} - 3.3\,\mathrm{km} - 1.1\,\mathrm{km}$) to 1:5 grid refinement ratio ($30\,\mathrm{km} - 6\,\mathrm{km} - 1.2\,\mathrm{km}$) yielded slightly worse WRF results (up to few percent difference in relative errors) in simulating the SEB components at our study sites.

Finally, we tested the sensitivity of our WRF simulations to the initialization setup. In addition to our original WRF runs, which have been initialized on day one of the simulation period and continuously run for the whole observational period (in addition to a 24-hour spin-up period), we performed separate WRF runs that were re-initialized each day (in addition to a 24-hour spin-up period each). Our results, as illustrated by the six-day example for simulated temperature (Figure 10), revealed that after the initial 36 hours, during which the two runs closely match each other, the runs differed by up to $4\,^\circ\mathrm{C}$ at a given hour, which led to a difference of up to $2.5\,^\circ\mathrm{C}$ (up to $40\,\%$ relative difference) for the daily-averaged temperature. A similarly large sensitivity to the choice of initialization procedure was found for the other meteorological variables analyzed in this study.

## 3.3 Evaluation of modeled melt energy

In simulating the daily-averaged $Q_M$ from the SEB model, the results from the reanalyses and WRF with the REF configuration benefit from the cancellation of biases between the overestimation of $K_{in}$ and the underestimation of $Q_H$ (Table 5, Figure S5). For the reanalyses, the overestimation of $K_{in}$ partly compensates for the underestimation of turbulent heat fluxes, resulting in a relatively good overall simulation of $Q_M$: underestimated with $6\,\%$ relative error for ERA5, and $10\,\%$ for ERA5-Land.

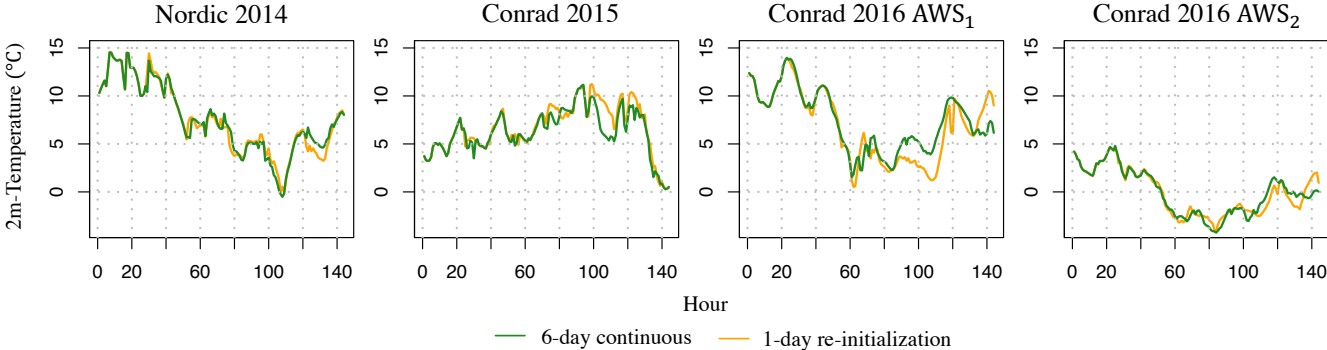

**Figure 10.** Timeseries of hourly temperature output from WRF at 1.1 km grid spacing derived from the six-day sensitivity runs based on the REF configuration. In green are the runs that are initialized on day 1 and continuously run for six days (in addition to a 24-hour spin-up period), while in yellow are the runs that are re-initialized per day and run for one day (in addition to a 24-hour spin-up period each). Here, temperature data is not lapse-rate bias corrected.

This compensation of biases between $K_{in}$ and $Q_H$ plays a lesser effect in the SEB model being forced by the WRF output: the partial cancellation of biases is only effective in the REF configuration as it is the only among the three configurations that overestimates $K_{in}$. Thus, REF yields the lowest relative error for $Q_M$ (underestimation of 5 % or WRF at 3.3 km and 8 % for
WRF at 1.1 km), followed by TOPSIS (underestimation of 15 % for WRF at 3.3 km and 19 % for WRF at 1.1 km ) and then minNRMSE (underestimation of 28 % for WRF at 3.3 km and 27 % for WRF at 1.1 km ).

Looking at the timeseries of daily $Q_M$, ERA5 and ERA5-Land are shown to closely resemble the observed timeseries, but fail at times to capture peak values of daily $Q_M$ (correlation $r_{sp}$ of 0.86, Figure 11). Across all sites, the SEB model forced by ERA5 and ERA5-Land yields stronger correlations between the modeled and observed timeseries of $Q_M$ than is the case
for WRF data ($r_{sp}$ of 0.72 for WRF at 3.3 km and 0.74 for WRF at 1.1 km with the REF configuration; Figure 11), but all the correlations remain statistically significant at 5 % confidence level.

While the model performance in simulating $Q_M$ is similar across the study sites, the model forced with reanalyses yields the best performance for Kaskawulsh glacier (nearly 0 % relative error for ERA5, and an overestimation by 1 % for ERA5-Land), while the worst performance is found for Nordic glacier ($Q_M$ underestimated by 15 % for ERA5 and 17 % for ERA5-Land).
When the model is forced with WRF data, the site with the best performance in $Q_M$ is Nordic glacier (overestimated by 3 % for WRF at 3.3 km and no relative error for WRF at 1.1 km with the REF configuration) and the site with the worst performance is the station in the accumulation zone on Conrad glacier in 2016 (underestimated by 18 % for WRF at 3.3 km and 16 % for WRF at 1.1 km). We find no dependence of the model performance to the size of glaciers in the sample nor to their geographical location; however, our sample size is too small to allow for a robust analysis of these relationships. Finally, we find negligible
differences between the SEB model performance when forced with the WRF at 3.3 km relative to 1.1 km (Table 5).

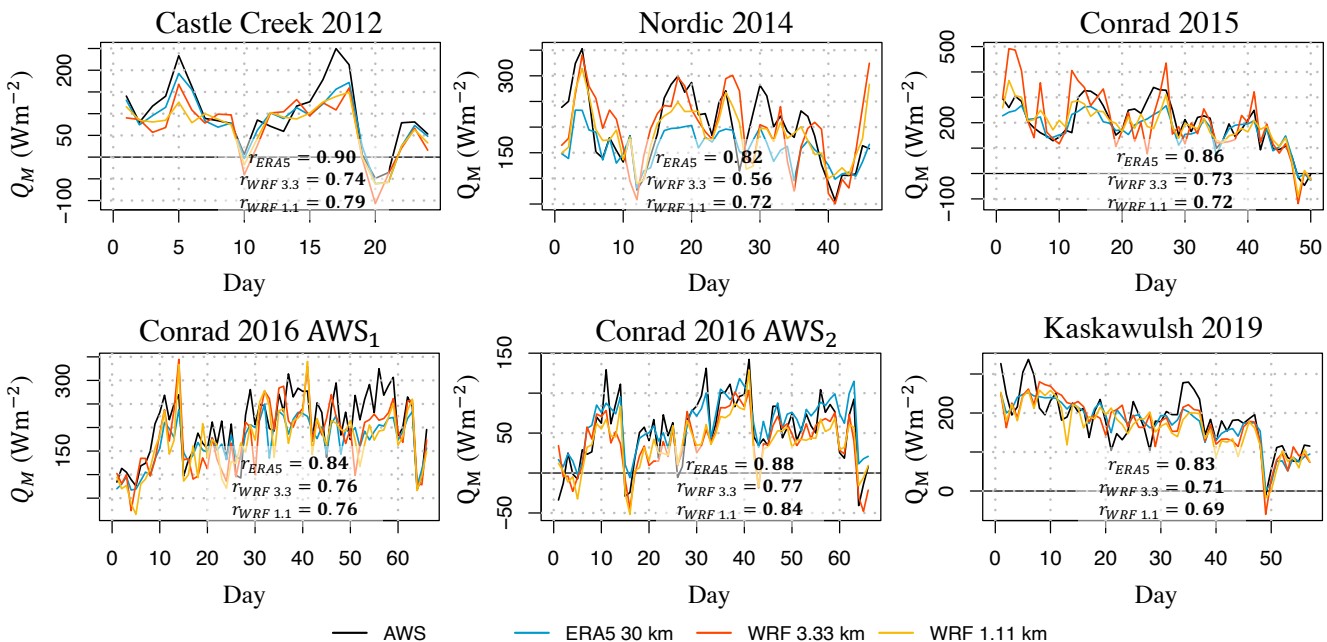

**Figure 11.** Modeled (ERA5, WRF at 3.3 km and WRF at 1.1 km) versus observed (AWS data) timeseries of daily melt energy ($Q_M$) over the observational period. Bold values of $r_{sp}$ indicate a statistically significant correlation at the 5 % confidence level. WRF is run with the REF configuration. The melt energy is estimated according to the SEB model (Eq. 1).

## 4 Discussion

In this study, we evaluated the use of ERA5 and ERA5-Land, with and without dynamical downscaling by WRF, in simulating meteorological variables needed to force a single-point SEB model at four mountain glaciers in Western Canada. We found that, with the exception of near-surface wind speed and relative humidity, all meteorological variables and energy fluxes are

similarly well simulated, based on NRMSE, by the reanalyses as well as by WRF at 3.3 km and 1.1 km. However, to adequately resolve near-surface temperature, the reanalysis and WRF needed to be lapse-rate bias corrected. The good performance of the reanalyses is, in part, expected since the reanalysis model incorporates data assimilation with available standard ground observations in the region as well as with remote sensing (Hersbach et al., 2020). Also, the good performance of ERA5 in simulating the daily variability of 2 m temperature, radiative fluxes and precipitation at our study sites corroborates previous

findings that focused on a more general evaluation of ERA5 across the globe and in this region. For example, an evaluation of ERA5 temperature and precipitation for hydrological modeling over Western Canada found that the model gives almost identical results when forced by observations and when forced by ERA5 (Tarek et al., 2020). Nevertheless, the study reported the largest difference between the observations and ERA5, in both temperature and precipitation, over mountainous terrain. Similarly, for different mountainous terrain across the globe, relatively large differences were found between observed and

ERA5 near-surface wind speed (Gualtieri, 2021), corroborating our findings of poorly simulated wind speed at our sites.

Our results reveal that the ERA5-Land reanalysis does not show a better performance compared to ERA5. In fact, we find that the performance of the SEB model forced by ERA5-Land is, on average, worse than when forced by ERA5 (10 % versus 6 % relative error for $Q_M$), though the differences are not statistically significant (Table S5). For comparison, we also looked into the output from WRF at 10 km grid spacing, i.e. at a similar grid spacing as in ERA5-Land (9 km). We found that, relative to ERA5-Land, WRF at 10 km performs better in simulating wind speed ($-24$ % versus $-74$ % relative error), incoming shortwave radiation ($-6$ % versus 11 % relative error), lapse-rate adjusted temperature ($-7$ % versus 23 % relative error) and relative humidity (2 % versus 28 % relative error). We note that ERA5-Land is produced without coupling to the atmospheric module and to the ocean wave models used by ERA5 (Muñoz Sabater et al., 2021), which might introduce biases in the variables analyzed in this study.

The downscaling of ERA5 with WRF has slightly improved the simulations of wind-speed and therefore turbulent heat fluxes as calculated from the bulk aerodynamic method in the SEB model. But more importantly, the WRF simulations of the SEB components at our sites are found to resemble the observations similarly well as the ERA5 does. In other words, the WRF model that is forced by ERA5 at its boundaries and is run without any data assimilation or 'nudging' to observations, performs similarly well to the state-of-the-art reanalysis model. These results are promising in terms of WRF application in downscaling long-term climate simulations from global climate models in order to project glacier evolution across this region.

While both reanalyses and WRF are found to simulate the daily melt energy at our sites similarly well, there are some differences in their simulations of key components of the SEB. Here we discuss these results in more detail.

(a) **Radiative fluxes.** Both reanalysis and WRF at 3.3 km and 1.1 km grid spacing, based on the REF configuration, over-estimate the frequency of clear sky days over the observational periods, which leads to overestimated mean incoming shortwave radiation ($K_{in}$) and to a lesser extent underestimated incoming longwave radiation ($L_{in}$) as an average over the observational period. While the local slope and shadow effects on $K_{in}$ are unlikely to be correctly captured in the reanalyses considering the coarseness (30 km and 9 km) of their native grid, capturing more successfully these effects in WRF did not result in improved simulations of $K_{in}$. In fact, at four of our six study sites, WRF at 1.1 km with the REF configuration showed the largest overestimation of $K_{in}$ among the datasets analyzed. These results corroborate previous findings arguing that the overestimation of downscaled $K_{in}$ indicates a problem in WRF with resolving convective clouds over complex terrain (e.g., Claremar et al., 2012; Collier et al., 2013). In the minNRMSE configuration, however, the number of clear sky days was underestimated, highlighting the sensitivity of results to the parameterization schemes used.

In addition to the choice of parameterization schemes, we investigated the impact on WRF output by switching the cumulus parameterization "on" and "off" in the innermost domains (3.3 km and 1.1 km), with "off" allowing for the cumulus convection to be resolved explicitly. We note that none of the cumulus schemes used in this study is scale-aware. Theoretically, cumulus parameterizations are only valid for coarse spatial grids of more than 10 km in order to release latent heat in the convective columns (Zhang et al., 2012). The parameterizations can also help to trigger mesoscale convection (5–10 km). For a grid spacing of 3–5 km or less, it is recommended to switch "off" the cumulus

schemes as the model can explicitly resolve deep convection and simulate convective storms (Skamarock and Klemp, 2008). However, it has also been recommended to keep this parameterization "on" for grid spacing of 1–10 km to avoid accumulated energy at grid points (Gerard, 2007). The cumulus parameterization scheme has been consistently turned "off" below 3 km in previous glacier studies (e.g., Mölg and Kaser, 2011; Collier et al., 2013, 2015; Aas et al., 2016). Between 3 and 5 km, some studies used the cumulus parameterization scheme (e.g., Mölg and Kaser, 2011), while others explicitly resolved deep convection without parameterization (e.g., Aas et al., 2016). Our results, in terms of the model performance in simulating $K_{in}$, show no systematic preference for either keeping the cumulus parameterization switched "on" or "off". This result may indicate that WRF's 1.1 km grid spacing might not be fine enough to correctly resolve the cumulus convection at our sites. Some studies recommended a grid spacing of the order of 100 m (Bryan et al., 2003; Petch, 2006) or even 10 m (Craig and Dörnbrack, 2008) to capture the dominant length scales of moist cumulus convection over complex terrain.

The commonly used land cover categories used for initializing WRF, based on the default MODIS data (Friedl and Sulla-Menashe, 2004) or ESA CCI data as used in this study (ESA, 2017, Table 2), do not distinguish between ice and snow categories. This distinction is crucial for the simulation of albedo on glacier surfaces, and, consequently, the net shortwave radiation. While WRF does simulate snowfall and therefore updates the surface albedo at each time step, the timeseries of modeled daily albedo can substantially differ from the in-situ observations (Figure 5), justifying our approach to use the observed daily albedo in the SEB model. Nevertheless, in the absence of observations, there are multiple albedo models of varying complexity (e.g., Oerlemans and Knap, 1998; Brock et al., 2000; Hirose and Marshall, 2013; Marshall and Miller, 2020) that could be incorporated in the SEB modeling, but this application is beyond the scope of our study. A promising result for these albedo models is that ERA5 and WRF timeseries of daily precipitation, including snowfall, are relatively well correlated with the timeseries of observed precipitation (Figure S2). This correlation analysis, however, may not be robust due to the likely poor quality of our in-situ precipitation measurements as highlighted before. More research is thus needed to adequately assess the performance of ERA5 and WRF for precipitation modeling at our sites.

(b) **Turbulent heat fluxes and temperature.** Previous work has found that ERA5 simulates high-quality surface turbulent fluxes across the globe (Martens et al., 2020), but the majority of reference observations in this evaluation came from stations in valleys and flat terrain, with few stations in the mountains and none from glacier surfaces. In our study, instead of directly taking the turbulent heat fluxes from the reanalysis and WRF, we calculated them using the bulk aerodynamic method with observed roughness lengths reported from previous studies at our sites (Table S3). Our observed roughness lengths agree with those estimated from other glaciers across the world (e.g., Denby and Smeets, 2000; Sicart et al., 2005). In the absence of any observations, a sufficient assumption would be to set the roughness length for momentum to $10^{-3}$ m, and the roughness lengths for temperature and humidity to $10^{-5}$ m. We note that WRF at 1.1 km grid spacing correctly represents the seasonally-averaged roughness lengths for momentum at our sites, while ERA5 does not (Table 4). With the roughness length correctly prescribed, the performance in simulating the turbulent fluxes in ERA5 and

WRF depends on how well their near-surface temperature and wind speed resemble those measured at the AWSs. We found that the lapse-rate bias corrections for temperature are necessary in order to provide more reliable estimates of turbulent heat fluxes, despite the substantial underestimation of wind speed in both the reanalyses and WRF. For example, without the lapse-rate corrections in ERA5, the relative error for daily mean temperature across all sites increased from the original 14 % to 54 %. We also showed that deriving the turbulent fluxes from the bulk method is preferable over taking the fluxes directly from the reanalyses and WRF.

(c) **Wind speed.** A study at large outlet glaciers ($> 100\,\text{km}^2$) and ice caps showed that WRF at a grid spacing of a few kilometers is able to successfully simulate katabatic winds (Claremar et al., 2012). However, the WRF model at 1.1 km fails to do so at our sites, including the large Kaskawulsh glacier. Choosing an appropriate grid spacing in WRF depends on the smallest weather features intended to be captured. While the smallest resolvable horizontal wavelength is twice the grid spacing, in practice the finite-difference equations used for advection and other dynamics in RCMs are unable to handle waves of this size, which either do not advect or are numerically unstable (Stull, 2015). Hence, these wavelengths are commonly filtered out of the models, and the smallest waves usually retained in RCMs are around five to seven times the grid spacing. Therefore, to be able to capture the local katabatic flow at our glacier sites, in theory we need a horizontal grid spacing smaller than one seventh of the glacier size (width and length). Even for Kaskawulsh glacier, the largest glacier in our study, this would require a grid spacing of well below 1 km. To test whether the performance in simulating wind speed improves at a sub-kilometer grid spacing, we ran WRF at 370 m grid spacing for a 16-day period only for the Kaskawulsh glacier site. For those runs, we use a high-resolution DEM with 30 m grid spacing (ASTER, 2019; Abrams et al., 2020). In order for these runs to be numerically stable, we also increased the number of vertical levels to 67 eta-levels, with a dense vertical layering in proximity of the surface (vertical spacing of 8 m in the first ∼60 m above the surface). Our results show that the simulation of wind speed is substantially improved, decreasing the original relative wind speed error of −77 % for WRF at 1.1 km for REF during the 16-day period (−81 % for minNRMSE, −77 % for TOPSIS) to a relative error of −14 % for REF (41 % for minNRMSE, −11 % for TOPSIS; Figure 12). The simulated wind direction also improved in these high-resolution WRF runs, yielding more resemblance with the downslope (katabatic) wind direction at the site (Figure 12). WRF at 370 m grid spacing also improved simulations of all other variables relative to WRF at 1.1 km, except for $L_{in}$ (Figure 13). However, the computational time of these high-resolution simulations increased by a factor of four relative to WRF at 1.1 km grid spacing, making simulations over a longer time frame challenging.

Apart from katabatic winds and synoptic storms, other meteorological phenomena mainly governed by topography, such as thermally-induced circulations and downslope windstorms, occur over mountain glaciers (Goger et al., 2022). Therefore, accurately representing the topography is crucial for correctly simulating the wind patterns. A better representation of topography explains the improved accuracy in wind speed and direction for smaller grid spacings in our simulations (1.1 km and 370 m). The finer grid spacing not only improves the elevation representation of the analyzed grid cell (Table S1), but also likely improves the elevation representation of the neighboring grid cells, leading to a more accurate

representation of slopes and aspects of the terrain. According to Wagner et al. (2014), the correct representation of topography is likely more important for the simulation of local flow regimes and turbulent heat fluxes than the choice of physics parameterization schemes.

**(d) Melt energy.** We found that the SEB model, when forced by the reanalyses and WRF, yields relatively small errors in simulated daily $Q_M$ relative to the SEB model being forced by the AWS data. However, this relatively small errors are in part due to the cancellation of biases between overestimated $K_{in}$ and underestimated $Q_H$ in the reanalyses and to a lesser effect in WRF. The underestimation of $Q_H$ down to −87 % in ERA5 and down to −64 % in WRF at 1.1 km is mainly due to underestimated near-surface wind speeds used in the bulk method. We note that the cancellation of biases is mainly reducing the mean bias error in $Q_M$ and not necessarily the variance error that evaluates the performance in simulating day-to-day variability. The strong and statistically significant correlations ($r_{sp} > 0.65$) between the modeled and 'observed' timeseries of daily $Q_M$ give additional confidence in the performance of both the reanalyses and WRF, but especially in ERA5 whose correlations from all the sites exceed 0.82 (Figure 9).

Testing the sensitivity of the WRF output to the choice of physics schemes revealed that the sensitivity is relatively small for the simulated surface melt energy across our study sites during the six-day periods (Figure 9). Over the whole observational period, however, the choice of different physics configurations leads to an underestimation of mean $Q_M$ by 8–27 % for WRF at 1.1 km with the differences in NRMSE in the range of 16–23 % (Table 5). Our sensitivity analysis also showed that for the individual components of the SEB, the choice of physics schemes can have a substantial impact on the simulations, with a difference of up to 50 % in NRMSE (Figure 9). This relatively high sensitivity corroborates findings in Wang et al. (2021) that the WRF model performance in glacierized and mountainous terrain strongly depends on the choice of physics parameterizations.

The performance of each physics scheme is likely dependent on the choice of schemes in other categories, which is a phenomenon that we did not investigate with our sensitivity tests where only one physics scheme was altered at a time. As a consequence, we did not sample the whole space of physics parameterization options and their combinations. However, from our relatively limited sampling, we did find that a WRF configuration that combines the best-performing schemes across the categories may not yield the best-performing simulations overall, i.e. across all variables in the SEB model. Similarly, the best-performing schemes overall, as identified by minNRMSE and TOPSIS, may not be the best-performing for each variable tested. To illustrate this example, we show the six-day simulations of $K_{in}$ from our 25 sensitivity runs, as well as from the minNRMSE and TOPSIS configurations (Figure 14). While the minNRMSE and TOPSIS configurations consist of the best-performing schemes according to the criteria used, neither of them yields the best performance in simulating $K_{in}$ among the configurations tested, and their simulations substantially differ from each other and the sensitivity runs (Figure 14).

To capture the characteristic synoptic time scales of four to seven days, our sensitivity tests were performed over a time period of six days, but the simulations are shown to be sensitive to the length of the time window and the timing during the melt season. For example, the TOPSIS run gave the best performance for the total melt energy in the sensitivity runs (six-day periods, four study sites), but not the best performance in stimulating $Q_M$ over the whole observational periods across all the sites. These results, as well as the sensitivity of WRF output to the initialization setup (Figure 10), indicate that rather

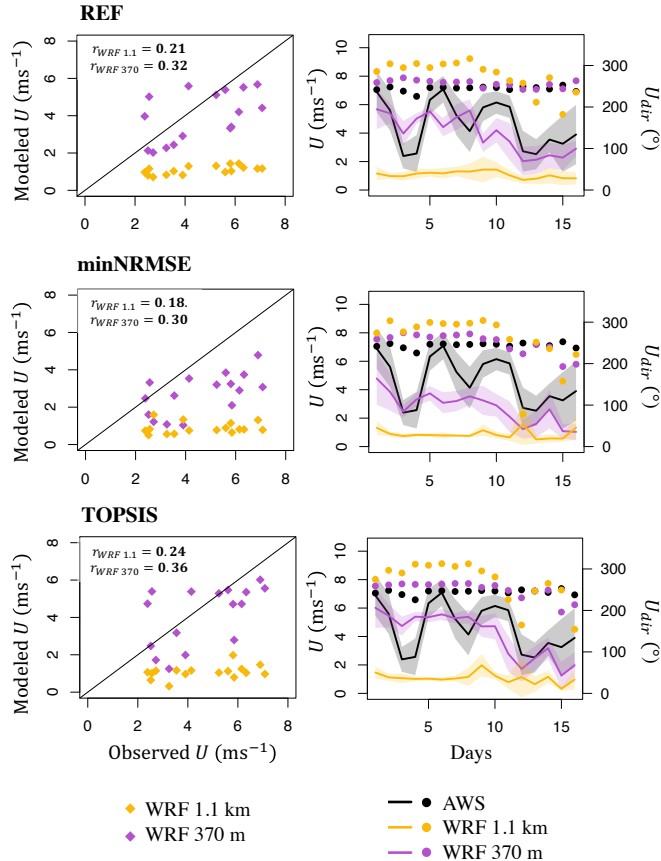

**Figure 12.** Left panel: Modeled versus observed (AWS) daily-averaged wind speed for WRF runs at 1.1 km (yellow) and 370 m (purple) grid spacing over a 16-day period for Kaskawulsh glacier site. WRF runs are based on three configurations: minNRMSE, TOPSIS, and REF. Right panel: Same results as above but shown as timeseries of daily wind speed ($U$, line), and daily wind direction ($U_{dir}$, dot). Bold values of $r_{sp}$ indicate a statistically significant correlation at the 5 % confidence level.

than settling with one "optimal" WRF configuration, it would be preferable to use an ensemble of WRF runs, each with a different configuration of physics parameterizations and initialization. For the application of WRF in downscaling long-term climate simulations, however, the use of ensemble runs can substantially increase the cost of already computationally expensive
simulations.

## 5 Conclusions

Our study aimed to address several knowledge gaps linked to the application of regional-scale physics-based models of glacier melt that require forcing by coarse-gridded data from reanalysis and global climate models. To address these gaps, in particular for glacier melt modeling in Western Canada, we asked: how well do the state-of-the-art reanalysis data, such as ERA5 and

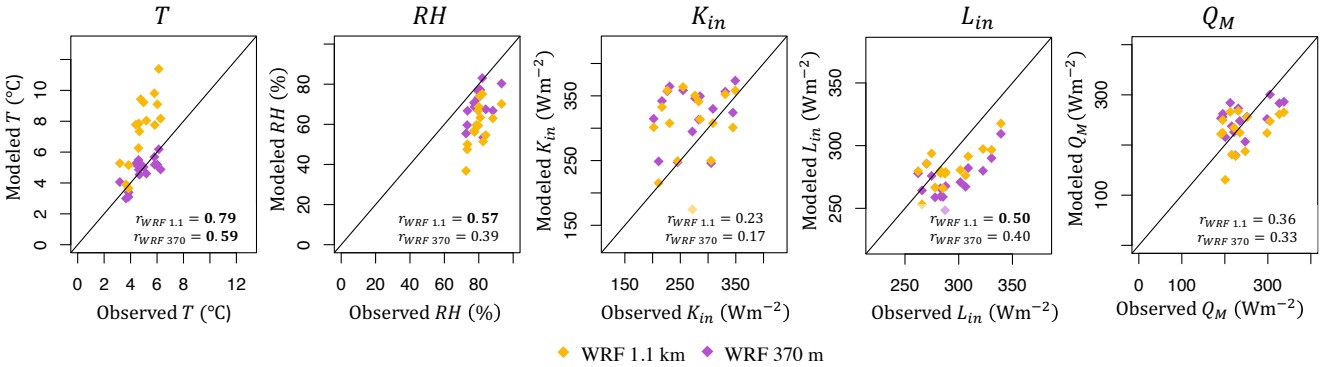

**Figure 13.** Modeled (WRF at 1.1 km and WRF at 370 m) versus observed (AWS) daily-averaged air temperature ($T$), relative humidity ($RH$), incoming shortwave ($K_{in}$) and longwave ($L_{in}$) radiation and total melt energy ($Q_M$) over a 16-day period for Kaskawulsh glacier site. Bold values of $r_{sp}$ indicate a statistically significant correlation at the 5 % confidence level. WRF is run with the REF configuration.

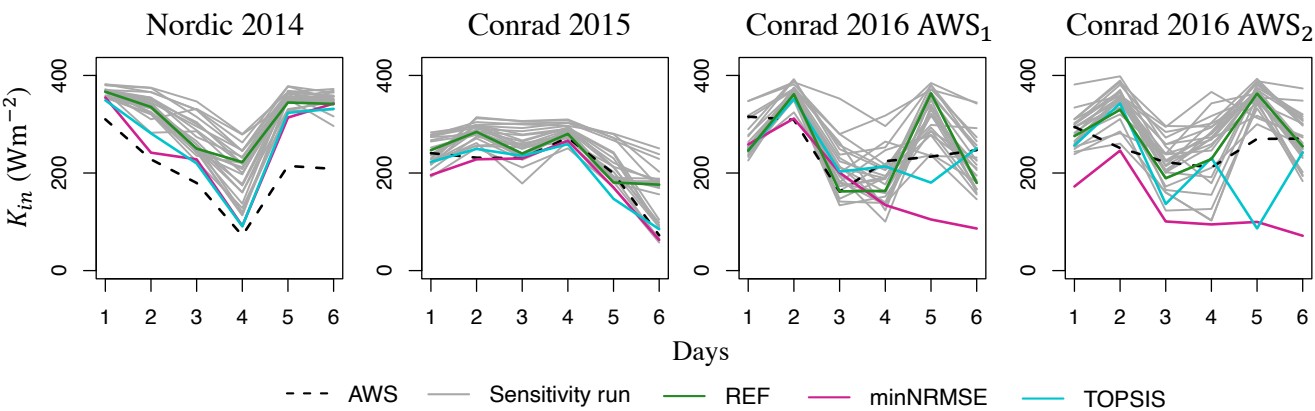

**Figure 14.** Timeseries of total daily incoming shortwave radiation ($K_{in}$) as observed at the four study sites (dotted black line), and as modeled according to the 25 WRF parameterization sensitivity runs (gray line), including the REF run (green line), and the two "optimal" runs: minNRMSE (pink line) and TOPSIS (blue line). All WRF runs are for 1.1 km grid spacing over a period of six days.

ERA5-Land, resemble the local-scale surface energy fluxes that drive melt at a glacier surface, and how well can these fluxes be resolved if ERA5 is dynamically downscaled with WRF to a scale of a few kilometers? To answer these questions, we focused on the four study glaciers in Western Canada with available in-situ measurements of all key components of SEB, collected by AWSs over different summer seasons in the period 2012–2019. To dynamically downscale ERA5, we used the WRF model with multiple nesting domains, and evaluated the WRF output at the two innermost domains at 3.3 km and 1.1 km grid spacing.

We also investigated the sensitivity of WRF output to its configuration and initialization, with a focus on the sensitivity to the choice of physics parameterization schemes.

We find that the mean melt energy over the observational periods is similarly well simulated (average underestimation of 6 %) when the SEB model is forced by ERA5 data and when it is forced by the AWS data, as long as the ERA5 temperature is lapse-rate bias corrected. The good performance of the reanalysis is also evidenced by the strong and statistically significant correlation ($r_{sp} > 0.82$) between timeseries of modeled and 'observed' daily melt energy at each study site. Relative to the observed fluxes at the sites, the mean radiative fluxes are well represented in ERA5, with 11 % average overestimation of incoming shortwave radiation, and no relative error for incoming longwave radiation. The sensible heat fluxes, on the other hand, are relatively poorly simulated (87 % average underestimation) mainly due to substantially underestimated near-surface wind speeds used to assess the fluxes via the bulk aerodynamic method. This overestimation of shortwave radiative fluxes and the underestimation of turbulent heat fluxes lead to a partial cancellation of biases in the modeled seasonal melt at each study site. Using ERA5-Land as input data, with a higher spatial resolution than ERA5, does not lead to improved simulations of surface energy fluxes.

Downscaling of ERA5 to 3.3 km and 1.1 km grid spacing with the WRF model improves the simulation of sensible heat fluxes (43–64 % average underestimation) and latent heat fluxes due to the improved simulations of wind speed, temperature and relative humidity, while the other fluxes remain similarly well simulated as in the reanalyses. As is the case with ERA5 data, but to a lesser extent, the SEB model forced with the WRF data benefits from partial cancellation of biases in the SEB components, leading to an average underestimation of 5–8 % for the mean melt energy at our study sites. However, these results depend on the WRF configuration, i.e. the set of physics parameterization schemes used in the model setup.

The sensitivity of WRF output to the choice of physics parameterizations is shown to be relatively low in simulating the total melt energy over the observational periods, but relatively high in simulating the individual components of the SEB. For our sites and observational periods, the parameterization schemes most commonly used in previous glacier studies with WRF application generally yield well performing simulations of surface energy fluxes among the configurations tested. These schemes include: Thompson microphysics scheme, Noah-MP land surface model, RRTMG shortwave and longwave radiation schemes, Grell 3D Ensemble cumulus scheme, and MYNN Level 3 scheme for planetary boundary layer and surface layer. The relatively high sensitivity of WRF results to the choice of parameterization schemes and initialization setup highlights the importance of ensemble WRF runs with different configurations rather than reliance on one "optimal" WRF configuration.

The WRF runs at 1.1 km grid spacing show similar or slightly worse results than those at 3.3 km grid spacing, but the differences are not statistically significant. The similarly successful performance of WRF and reanalysis, as input to the SEB model at our glacier sites, increases the confidence in using ERA5 for reconstructions of past glacier melt in this region, as well as in using WRF for downscaling simulations from global climate models to derive long-term projections of glacier melt. The use of physics-based melt models, forced with reliably downscaled fields from global climate models, is a path toward narrowing the uncertainties in projections of glacier contribution to streamflow and sea level rise.

*Code and data availability.* This study is based on the output from the Weather Research and Forecasting (WRF) model, version 4.1.3, available for download at https://github.com/wrf-model/WRF. ERA5 reanalysis data is available online from the Copernicus Climate Data

Store (https://cds.climate.copernicus.eu). The AWS data from the study glaciers are part of published articles (Radić et al., 2017; Fitzpatrick et al., 2017, 2019; Lord-May and Radić, 2023), and are available upon request from the corresponding authors of these studies.

*Author contributions.* Data collection and analysis were performed by Christina Draeger under supervision of Valentina Radić. Mekdes Tessema helped inform the analysis with earlier WRF runs. The first draft of the manuscript was written by Christina Draeger and all authors commented on previous versions of the manuscript. All authors read and approved the final manuscript.

*Competing interests.* The authors declare no competing interests.

*Acknowledgements.* Funding supporting this study was provided through the Natural Sciences and Engineering Research Council (NSERC) of Canada (Discovery grants to Valentina Radić, Rachel H. White). Our meteorological equipment is supported by a NSERC Research Tools and Instruments grant and a Canada Foundation for Innovation grant (Valentina Radić).

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
