# Peer review of "Evaluation of ERA5 and Dynamical Downscaling for Surface Energy Balance Modeling at Mountain Glaciers in Western Canada"

_EGUsphere, 2023_

## Author Comment (AC1)

We thank both reviewers for their useful comments and positive feedback. The suggested revisions substantially improved the manuscript, and we addressed all comments, point-by-point, in this document. The comments of the reviewers are shown in black and our replies in blue. We number reviewer comments for referencing purposes throughout the document (comment 1 = C1, etc.). The line, figure and table numbers are based on the updated manuscript.

**Responses to Referee #1 (R1)**

**Summary**

This manuscript presents an overview over the performance of ERA5, ERA5-LAND and dynamical downscaling with the WRF model (to dx~3km and dx~1km) for calculating the surface energy balance (SEB) of mountain glaciers over Western Canada. Four glaciers are chosen for evaluation, where observations of the relevant variables (e.g., turbulent heat fluxes, temperature, radiation, wind speeds, etc.) are available during the summer season. The authors derive the simulated variables for the SEB, after some corrections, from the model output and directly compare the results with the observed values. Furthermore, they run the WRF model in multiple configurations for parametrizations to find the "optimal" setup for a satisfactory calculation of the SEB. Results suggest that dynamical downscaling with WRF does not automatically outperform ERA5, except for the wind speed and direction - mostly due to the higher horizontal resolution. Generally speaking, both ERA5 and WRF are useful for calculating the SEB, while a correct simulation of the meteorological fields over the glaciers would require even higher horizontal resolution at the hectometric range.

The manuscript is extensive and has a valuable purpose in discussing the challenges of dynamical downscaling over glaciated environments and suggesting an "optimal" setup for future applications. However, in some sections, the authors need to argue in more detail on why they apply a new method; some reasonings are given in the discussion, while they would be already required in the methods section. The interpretation the WRF results is sometimes lacking an important factor - namely terrain resolution. Comments and suggestions are given in the list below.

We thank the reviewer for their useful comments and positive feedback that helped us improve the quality of the manuscript. By addressing these comments, we have been able to provide a clearer explanation of the need for a new method and enhance the discussion of key points.

**Major comments**

*R1 C1:* **Calculation of the surface fluxes from model output via the bulk method**. I agree that the modelled albedo values strongly differ from the observations; however, while only reading the methods it is difficult to follow the argumentation why the authors decide to calculate the turbulent fluxes with the observed albedo via the bulk method instead of directly using the values for sensible & latent heat fluxes from model output. Is this a common method to utilize output from an atmospheric model for glacier SEB modelling- was this approach also used in previous studies?

We thank the referee for this comment, and we realized that more clarification on this issue should be introduced from the start (in the "Data and Methods" section) rather than in the "Discussion" section. As we see it, there are two points raised by the referee: one is the use of observed albedo rather than derived albedo from ERA5 and WRF, and the other is the use of the commonly used bulk methods (in glacier studies) to derive turbulent heat fluxes rather than outputting these directly from ERA5 and WRF. (Please note that albedo does not feature in the calculation of turbulent heat fluxes, but in the calculation

of net shortwave radiative fluxes – we guess that the referee points to the fact that albedo will be linked to roughness length, i.e. depending whether snow or ice albedo is on the surface the roughness lengths are determined.) Both albedo and turbulent fluxes, as daily timeseries, are known to be poorly simulated by ERA5 and WRF at relatively coarse spatial resolution (> 1 km grid spacing). We also show this poor simulation in the "Discussion" section. Prompted but the referee's comment, we now provide more rationale in the "Data and Methods" section to justify our use of the observed albedo and the bulk method (lines 291-308 in the updated manuscript). For accurately modeling turbulent fluxes, it is crucial that both ERA5 and WRF correctly represent the glacier surface roughness lengths, as well as the temperature, humidity, and wind speed in the surface boundary layer (SBL). Considering that ERA5 and WRF fail to do so at their respective grid spacing, their derived turbulent heat fluxes (regardless of the parametrization scheme used for the SBL) are far off from the observed values. We also added Figure 5 and Table 4 to the main text, representing the modeled vs. observed timeseries of daily albedo and seasonal roughness lengths, respectively.

*Lines 291-308: The primary goal of the evaluation analysis is to assess the performance of the SEB model, forced with either ERA5 or WRF data, in simulating seasonal melt energy at our sites. To do so we evaluate the total simulated energy available for melt ($Q_M$ ; Eq. 1), as well as the daily timeseries of $Q_M$ , as calculated from the SEB model forced with the reanalyses (ERA5, ERA5-Land), as well as with the WRF output, against the reference calculations when the same SEB model is forced with the AWS data. Thus, the input for the SEB model, i.e. the atmospheric variables $K_{in}$, $L_{in}$, $T$, $RH$, and $U$, are taken from: (1) the AWS at each site, representing the reference or true values, (2) ERA5, (3) ERA5-Land, and (4) WRF at grid spacings of 3.3 and 1.1 km, using each of the three configurations (REF, min-NRMSE, and TOPSIS). For the reanalysis and WRF, only the data from the grid cell covering each study site is used. As we are interested in the evaluation of meteorological rather than surface variables (albedo and surface roughness), we use in-situ observations of daily surface albedo and seasonally-averaged roughness lengths in the SEB model. These surface variables could have been taken directly from the reanalysis and WRF; however, we found that these values can differ substantially from the observed ones throughout the observational periods (Figure 5 and Table 4). The discrepancy between WRF and observed albedo at glaciers, especially in the ablation zone, has also been noted previously in Eidhammer et al. (2021). Thus, to avoid any evaluation biases originating in poorly assigned surface variables, we stick to the choice of using observed surface variables in the SEB model. The observed daily surface albedo is calculated as the ratio of measured daily totals (in local daylight hours) of reflected and incoming shortwave radiation at each site. The incoming shortwave radiation at the surface is taken from these datasets without any further modifications (e.g. separation into direct and diffuse radiation).*

**Table 4.** Mean seasonal roughness lengths [m] for momentum ($z_{0v}$) derived from the observations (AWS), ERA5 and WRF (1.1 km) at each study site.

| Site | AWS | ERA5 (30 km) | WRF (1.1 km) |
|------|-----|--------------|--------------|
| Castle Creek 2012 | 0.003 | 0.067 | 0.002 |
| Nordic 2014 | 0.003 | 0.936 | 0.002 |
| Conrad 2015 | 0.003 | 1.169 | 0.002 |
| Conrad 2016 AWS$_1$ | 0.001 | 1.166 | 0.002 |
| Conrad 2016 AWS$_2$ | 0.003 | 1.166 | 0.002 |
| Kaskawulsh 2019 | 0.001 | 0.001 | 0.002 |

[Figure]

**Figure 5.** Modeled (ERA5, WRF at 3.3 km and WRF at 1.1 km) versus observed (AWS data) timeseries of daily albedo over the observational period (starting at Day 1) at each site. WRF is run with the REF configuration.

The authors mention in the discussion the unsatifactory performance from the turbulent fluxes from the direct model output (lines 497--508), but for the general understanding of the manuscript, it would make sense to add these paragraphs directly after they introduce the new method (ca. Line 385).

As requested, we now moved the discussion of the performance from the turbulent fluxes from the direct model output from the discussion to the "Results" section. We also provided more detailed rationale in the "Data and Methods" section (lines 291-308; see comment R1 C1).

Furthermore, changing one parameter to derive a quantity from the rest of the modelled output might lead to physical inconsistencies, because all the other variables used for the bulk method still indirectly depend on the "wrong" albedo. Did the authors calculate the SEB with directly modelled turbulent fluxes?

It is correct that we use the bias-corrected temperature, humidity, and wind speed directly from WRF/ERA5, while surface variables (albedo and roughness lengths) are derived from observations. The referee is correct that this leads to physical inconsistency in the use of modelled simulations; however the main objective of this study was to investigate how well an 'off-grid' SEB model (meaning that the SEB model is decoupled from WRF and ERA5 models) performs at a given location on a glacier, when forced with meteorological variables from WRF/ERA5 in comparison to when forced with AWS data. We now make this objective more clear and also mention the limitations associated with this approach (lines 297-317, see comment R1 C1).

Yes, we also evaluated the performance of directly modeled turbulent fluxes from WRF/ERA5 in the SEB model and this resulted in large biases relative to the SEB model when forced with AWS data. We had previously addressed this in the "Discussion" section, but to highlight this finding, we have moved it to the "Results" section (lines 403-413):

*We also investigated the use of surface $Q_H$ and $Q_L$ as outputted directly from the reanalysis and WRF into the SEB model, rather than calculating those fluxes with our bulk method. In WRF, these fluxes are derived through a local or non-local closure scheme in the planetary boundary and surface layer, depending on the parameterizations used (Skamarock and Klemp, 2008). When $Q_H$ is directly taken from ERA5, the NRMSE of $Q_H$ is 83 %, which is twice as large as the original error when $Q_H$ is calculated with the bulk method. In WRF at 1.1 km, the error in $Q_H$ is increased from 31 % when the bulk method is used to 60 %, while the error for $Q_L$ is increased from 21 % to 54 %. For Kaskawulsh glacier, the largest glacier among our study sites, the performance of simulated $Q_H$ and $Q_L$ directly from ERA5 is similar or only slightly worse (few percent) than the performance based on the bulk method. However, looking across all the sites, taking $Q_H$ and $Q_L$ directly from ERA5 leads to an increased underestimation of mean $Q_M$ from 6% in the original estimate to 72%. For WRF at 1.1km, the relative error in $Q_M$ increased from 8% in the original estimate to 17%. These results justify our choice to assess the turbulent heat fluxes via the bulk method instead of taking them directly from the reanalyses and WRF.*

*R1 C2:* **Interpretation - terrain resolution**. The authors argue that the poor performance of wind speed and direction simulation yields from the inability to simulate the katabatic glacier wind. The authors could check whether the "bad" model performance only happens during the wind directions corresponding to the down-glacier wind - the model seems to perform better during synoptically-forced conditions. However, glacier winds are not the only meteorological phenomenon present over mountain glaciers; such as thermally-induced circulations, downslope windstorms, etc, which are all mostly governed by the topography (Goger et al, 2022). Therefore, well-resolved topography is essential for the correct simulation of the wind field - tis also explains the general bias reduction of wind speed & direction for small horizontal grid spacings (dx=1.1km and dx=370m). This is an important point which should be mentioned in the discussion and interpretation of the results. Publications from idealized simulations argue that at least 10 points across a valley are necessary to simulate the relevant processes well, and that the correct representation of topography is likely more important than the choice of parameterization schemes (Wagner et al, 2014).

We thank the reviewer for this comment. We incorporated a more extensive discussion on this topic in the paragraph on wind speed in the discussion (lines 586-594):

*Apart from katabatic winds and synoptic storms, other meteorological phenomena mainly governed by topography, such as thermally-induced circulations and downslope windstorms, occur over mountain glaciers (Goger et al., 2022). Therefore, accurately representing the topography is crucial for correctly simulating the wind patterns. A better representation of topography explains the improved accuracy in wind speed and direction for smaller grid spacings in our simulations (1.1 km and 370 m). The finer grid spacing not only improves the elevation representation of the analyzed grid cell (Table S1), but also likely improves the elevation representation of the neighboring grid cells, leading to a more accurate representation of slopes and aspects of the terrain. According to Wagner et al. (2014), the correct representation of topography is likely more important for the simulation of local flow regimes and turbulent heat fluxes than the choice of physics parameterization schemes.*

*R1 C3:* **TOPSIS and minRMSE configurations.** Maybe I missed it, but do the authors somewhere list the final WRF model setup of TOPSIS and minRMSE, like Table 2 for the REF run? This might be of use for future dynamical downscaling studies.

Yes, the final WRF setup for TOPSIS and minNRMSE was listed in the Supplementary Information. We now moved this table to the main text (Tables 2 and 3).

**Minor comments**

*R1 C4:* line 50: which simplified assumptions?

We revised the sentence (lines 52-54):

*Nevertheless, as statistical downscaling relies on simplified assumptions (e.g., the existence of linear relationships between local and large-scale climate variables), the technique introduces another source of error or uncertainty into the model output (Marzeion et al., 2020).*

*R1 C5:* line 57: make a new paragraph

Done.

*R1 C5:* line 83: An extensive analysis of real-case, high-resolution large-eddy simulations over a glacier is provided by Goger et al (2022), and Sauter & Galos (2016) performed semi-idealized LES over a glacier and evaluated the calculation of turbulent fluxes.

Thank you for pointing out these studies. We updated the references in the text (lines 88-90).

*R1 C6:* line 85: "Downscaling to several kilometers": Several kilometers might not be the optimal target for mountain glaciers embedded in highly complex terrain, which requires likely horzintal grid spacings of less than 1km.

We agree with the reviewer and revised the sentence (lines 90-92):

*Therefore, when incorporating WRF into long-term glacier evolution modeling at regional scales, downscaling to a grid spacing of approximately one kilometer seems to be the computationally optimal target.*

*R1 C7:* lines 134-203: I understand that it is important to mention the most commonly used parameterization schemes in WRF, but this is too lengthy for an introduction - perhaps it's enough to mention this configuration in the methods and finally say how it performs within the ensemble.

We have condensed the introduction to focus only on the challenge of identifying optimal physics parameterization schemes in WRF (lines 93-102). The detailed explanation of commonly used parameterization schemes in glacier studies has been moved to the "Data and Methods" section (209-221).

*R1 C8:* line 213: You can place the optimal configuration of parameterizations from the introduction here. Done accordingly.

*R1 C9:* line 220: What do you mean exactly by "reflect different time windows during melt season"?

We randomly assigned the six-day periods for the sensitivity analysis to represent different time periods in the melt season (e.g. early, middle and late melt season). The sentence is revised accordingly (lines 229-230):

*The six-day periods are selected randomly to represent different time windows throughout the early, middle, and late melt season.*

*R1 C10:* line 398: " none of these altered WRF configurations yield a strong impact on the calculated Q_M from the SEB model": Did you reset the albedo for calculating the turbulent fluxes here as well? Because then this relative agreement is not very surprising.

The albedo and roughness lengths are set to the observed values and are kept the same across all three WRF configurations. Considering that the simulation of individual components of the SEB components, such as the radiative and turbulent fluxes, can vary substantially among the three WRF configurations (as observed in the sensitivity tests, Figure 14), the relative agreement in simulating $QM$ is somewhat surprising. Despite the substantial differences in these individual SEB components due to different parameterization choices, the simulation of melt energy over the observational period remains relatively consistent for the three optimal configurations.

*R1 C11:* line 478: ...."do not distinguish between ice and snow categories": It's true that the land use category does not distinguish between snow and ice. However, after initialization, WRF indeed initializes snow cover on glacierized surfaces. The authors mention observed snow cover at one of the glaciers during the time window of interest - is this snow cover present in WRF as well? If yes, the snow cover indeed has an influence on the SEB in the model.

Here in the text, we were referring to the initial land category data. Yes, WRF does simulate the snowfall and therefore updates the snow cover in the simulations which is then reflected in the surface albedo for each day/hour. We now made this clear in the text (lines 534-546). We did compare the timeseries of daily albedo from WRF (and ERA5) with the observed albedo at our sites and found large discrepancies (lines 300-306, see R1 C1; Figure 5). This figure has now been moved to the main text to better motivate our choice of using the observed albedo in the SEB model.

Also, in response to comment R2 C1 from referee #2, we have included a comparison between modeled (WRF and ERA5) and observed daily precipitation at our research sites (lines 355-362 and 374-379, Figures S2 and S3). Although precipitation (in the form of rainfall) contributes relatively little to the SEB at seasonal scales, events of fresh snowfall can significantly impact albedo and consequently the melt energy, depending on the frequency of such events during a melt season.

*Lines 534-546: The commonly used land cover categories used for initializing WRF, based on the default MODIS data (Friedl and Sulla-Menashe, 2004) or ESA CCI data as used in this study (ESA, 2017, Table 2), do not distinguish between ice and snow categories. This distinction is crucial for the performance of albedo on glacier surfaces, and, consequently, the net shortwave radiation. While WRF does simulate snowfall and therefore updates the surface albedo at each time step, the timeseries of modeled daily albedo can substantially differ from the in-situ observations (Figure 5), justifying our approach to use the observed daily albedo in the SEB model. Nevertheless, in the absence of observations, there are multiple albedo models of varying complexity (e.g., Oerlemans and Knap, 1998; Brock et al., 2000; Hirose and Marshall, 2013; Marshall and Miller, 2020) that could be incorporated in the SEB modeling, but this application is beyond the scope of our study. A promising result for these albedo models is that ERA5 and WRF timeseries of daily precipitation, including snowfall, are relatively well correlated with the observed timeseries (Figure S3). This correlation analysis, however, may not be robust due to the likely poor quality of in-situ precipitation measurements as highlighted before. More research is thus needed to adequately assess the performance of ERA5 and WRF in precipitation modeling at our sites.*

*R1 C12:* Figures 8 and 10: Please add a background grid to the figure, this improves their readability.

Done.

**References**

Goger, B., Stiperski, I., Nicholson, L., and Sauter, T. (2022): Large-eddy simulations of the atmospheric boundary layer over an Alpine glacier: Impact of synoptic flow direction and governing processes, Q. J. R. Meteorol. Soc, 148, 1319–1343, https://doi.org/10.1002/qj.4263

Sauter, T. and Galos, S. P. (2016): Effects of local advection on the spatial sensible heat flux variation on a mountain glacier, The Cryosphere, 10, 2887–2905, https://doi.org/10.5194/tc-10-2887-2016

Wagner, J. S., A. Gohm, and M. W. Rotach (2014): The impact of horizontal model grid resolution on the boundary layer structure over an idealized valley. Mon. Wea. Rev., 142, 3446–3465, https://doi.org/10.1175/MWR-D-14-00002.1

---

## Author Comment (AC2)

We thank both reviewers for their useful comments and positive feedback. The suggested revisions substantially improved the manuscript, and we addressed all comments, point-by-point, in this document. The comments of the reviewers are shown in black and our replies in blue. We number reviewer comments for referencing purposes throughout the document (comment 1 = C1, etc.). The line, figure and table numbers are based on the updated manuscript.

**Responses to Referee #2**

This paper investigates the use of ERA5 reanalysis data set, with and without dynamical downscaling to evaluate surface energy balance modeling of glaciers in western Canada. The authors look specifically at the variables that are used to calculate the energy available for melt when evaluating the different forcing sets. For downscaling they use the Weather Research Forecasting (WRF) model, and include several tests with various parametrization combinations in WRF and try to determine the best combination for the western region in Canada. I believe this paper is a nice contribution to anyone interested in glacier mass balance studies in this and similar regions and should be published after addressing the comments and suggestions.

We thank the reviewer for their useful comments and positive feedback that helped us improve the quality of the manuscript. By addressing these comments, we were able to improve the discussion of some crucial points.

**Major Comment**

*R2 C1:* Page 12, line 257. Does the Fitzpatrick 2019 paper actually support neglection of heat flux from rain (or ground?) The only thing I could find was that "while readings from periods affected by precipitation on the analyser windows were removed" for eddy covariance data. That does not warrant negligible contribution from rain. I suggest removing reference to the 2019 paper. However, the Fitzpatrick 2017 paper is a nice citation. And I found this paragraph interesting: "QR provided a negligible contribution (<1%) to the total melt energy over the recorded period. However, over daily and sub-daily timescales, QR was observed to have a considerable influence on SEB and ablation during heavy rainfall."
Precipitation is one of the outputs you want to get correct in WRF. Though it might not have been important for the study window in this manuscript, I do believe a WRF parameterization combination that gets the precipitation correct (along with other metrics) is desired (and as stated in the Fitzpatrick 2017 paper, "QR was observed to have a considerable influence on SEB and ablation during heavy rainfall"). I would have liked to see which parameter combinations score the best when precipitation also is included.

We thank the reviewer for this comment. In response, we have included more references justifying the omission of rain flux and ground heat flux from our analysis, as these factors have been shown to have negligible contributions to the seasonal melt energy. We now highlight that we are primarily interested in the performance of SEB modeling for seasonal melt simulations and in exploring the suitability of ERA5 for long-term glacier melt simulations, both with and without downscaling. We agree with the reviewer that precipitation is an important variable, especially for the simulation of accumulation and therefore the glacier-wide mass balance.

Although precipitation (as rainfall) can significantly contribute to the melt energy on specific days during a melt season, as shown in Fitzpatrick et al. (2017), uncertainties associated with assessing the rain heat flux in models are relatively large and potentially grounded in unsupported assumptions (Hock, 2005; Fitzpatrick et al., 2017). However, more important for SEB modeling is precipitation in the form of

snowfall, which can substantially alter the albedo and therefore the melt energy depending on the frequency of these fresh snowfall events during a melt season (e.g., Hock, 2005; Mac Dougall et al., 2011; Marshall and Miller, 2020).

We now elaborate on these limitations in more detail in the "Data and Methods" section when justifying the selection of our SEB model (lines 264-270). Following the reviewer's suggestion, we have included a comparison between modeled (from WRF and ERA5) and observed daily precipitation at our research sites (lines 355-362 and 374-379, Figures S2 and S3). We also updated Tables 5, S4 and S5 to include precipitation.

*Lines 264-270: Given our focus on the key seasonal SEB components, we neglect the ground heat flux and the heat flux from rain, since both have been shown to give negligible contributions to the total seasonal melt at mid-latitude glaciers (Sicart et al., 2005; Andreassen et al., 2008; Gillett and Cullen, 2011), as well as at our study sites (Fitzpatrick et al., 2017; Fitzpatrick, 2018; Lord-May and Radić, 2023). The rain heat flux, however, can be a substantial contributor (up to 20 %) to daily melt energy on a day with extreme rainfall (Fitzpatrick et al., 2017), but the uncertainty in the model used to assess the rain heat flux is relatively large (Hock, 2005; Fitzpatrick et al., 2017). For these reasons, we neglect the heat flux in the SEB model, but will include precipitation, both as rainfall and snowfall, in the evaluation analysis.*

*Lines 355-362: Seasonally-averaged total daily precipitation (P) from ERA5 is overestimated at some glacier sites (relative error of 130.3 % for Kaskawulsh 2019 and 84.5 % for Conrad 2016 AWS1), while underestimated at other sites (relative error of –37.9 % for Conrad 2016 AWS2, –33.2 % for Conrad 2015, and –18.7 % for Nordic 2014; Figures S2 and S3). On the other hand, the modeled timeseries of daily precipitation all show statistically significant correlation (p-value < 0.05) with the observed timeseries (Figure S2). The frequency of days with heavy snowfall is overestimated in the ablation zones (Castle Creek 2012, Kaskawulsh 2019) and underestimated in the accumulation zone (Conrad 2016 AWS2). On average, ERA5-Land precipitation simulations perform worse than ERA5 (mean overestimation of 35.6 % in ERA5-Land vs. 20.3 % in ERA5 across all sites).*

*Lines 374-379: The modeled timeseries of daily precipitation shows statistically significant correlation (p-value < 0.05) with the observed timeseries for all sites except for Kaskawulsh glacier (Figure S2). WRF tends to overestimate the frequency of days with heavy snowfall in both the ablation zone (Castle Creek 2012, Kaskawulsh 2019) and the accumulation zone (Conrad 2016 AWS2; Figure S3). Additionally, it also overestimates the frequency of days with light rainfall (< 2.5 mm). On average, precipitation values from WRF at 3.3 km perform significantly worse than from WRF at 1.1 km (29.0 % vs. –5.0 % relative error with the REF configuration; Table 5).*

[Figure]

**Figure S2:** Modeled (ERA5, WRF at 3.3 km and WRF at 1.1 km) versus observed (AWS data) timeseries of daily total precipitation over the observational period. Bold values of $r_{sp}$ indicate a statistically significant correlation at the 5 % confidence level. WRF is run with the REF configuration.

[Figure]

**Figure S3:** Modeled (ERA5, ERA5-Land, WRF at 3.3 km and WRF at 1.1 km) versus observed (AWS data) densities of (a) total, (b) liquid and (c) frozen daily averages of precipitation over the observational period. We differentiate between liquid and frozen precipitation using a temperature threshold of 0 °C. The densities of liquid and frozen precipitation are displayed only if there were days with snowfall during the observational period. WRF is run with REF parameterization schemes.

**Table 5.** Relative difference (%) between modeled and observed seasonally-averaged values, as well as NRMSE between modeled and observed daily values of: air temperature ($T$), relative humidity ($RH$), total precipitation ($P$), wind speed ($U$), incoming shortwave ($K_{in}$) and longwave ($L_{in}$) radiation, sensible ($Q_H$) and latent ($Q_L$) heat fluxes, and total melt energy ($Q_M$). The melt energy is estimated according to the SEB model (Eq. 1). The WRF runs are based on the three configurations of physics parameterizations: REF, minNRMSE and TOPSIS. For comparison, we also include the results of the ensemble-mean, where $T$, $RH$, $P$, $U$, $K_{in}$ and $L_{in}$ are derived as a mean across the three configurations. The turbulent heat fluxes and $Q_M$ in the ensemble-mean are derived according to the aerodynamic bulk method (Eqs. 2 and 3) and SEB model (Eq. 1), respectively. The results of each evaluation metric are shown as the mean ($\pm$ one standard deviation) across the six study sites, with equal weighing of each site. For seasonally-averaged values of $Q_M$, we only take into account positive values of $Q_M$ that drive melt. Values in bold highlight the best performing model for the given variable according to the metric used.

| Variable | ERA5 30 km | ERA5-Land 9 km | WRF 3.3 km REF | WRF 3.3 km minNRMSE | WRF 3.3 km TOPSIS | WRF 3.3 km Ensemble | WRF 1.1 km REF | WRF 1.1 km minNRMSE | WRF 1.1 km TOPSIS | WRF 1.1 km Ensemble |
|---|---|---|---|---|---|---|---|---|---|---|
| | | | | | | | | | | |
| **Relative Error (%)** | | | | | | | | | | |
| **T** | 14 ± 24 | 23 ± 23 | **3** ± 37 | -21 ± 48 | 9 ± 37 | **-3** ± 40 | 6 ± 30 | -25 ± 39 | 11 ± 29 | **-3** ± 32 |
| **RH** | 30 ± 12 | 28 ± 12 | **-2** ± 14 | 17 ± 22 | -6 ± 12 | 3 ± 14 | -9 ± 11 | 10 ± 16 | -12 ± 9 | -4 ± 12 |
| **P** | 20 ± 70 | 36 ± 99 | 29 ± 88 | 46 ± 101 | **1** ± 45 | 25 ± 77 | -5 ± 51 | 12 ± 53 | -15 ± 27 | -3 ± 43 |
| **U** | -64 ± 10 | -74 ± 6 | -26 ± 18 | -42 ± 12 | **-22** ± 20 | -30 ± 15 | -44 ± 16 | -46 ± 14 | -42 ± 15 | -44 ± 14 |
| **K$_{in}$** | 11 ± 8 | 11 ± 8 | 12 ± 6 | -16 ± 16 | -4 ± 9 | -3 ± 9 | 19 ± 16 | -11 ± 18 | **1** ± 11 | 3 ± 14 |
| **L$_{in}$** | **0** ± 4 | -1 ± 4 | -5 ± 3 | **0** ± 4 | -3 ± 3 | -3 ± 3 | -5 ± 3 | -1 ± 3 | -3 ± 2 | -3 ± 2 |
| **Q$_H$** | -87 ± 11 | -95 ± 5 | -43 ± 38 | -81 ± 14 | **-40** ± 40 | -58 ± 22 | -64 ± 16 | -82 ± 13 | -64 ± 16 | -73 ± 13 |
| **Q$_L$** | 520 ± 977 | **113** ± 335 | -613 ± 1203 | -115 ± 207 | -804 ± 1533 | -399 ± 758 | -383 ± 453 | -235 ± 298 | -531 ± 702 | -368 ± 437 |
| **Q$_M$** | -6 ± 7 | -10 ± 7 | **-5** ± 10 | -28 ± 10 | -15 ± 15 | -26 ± 16 | -8 ± 7 | -27 ± 10 | -19 ± 8 | -29 ± 15 |
| **NRMSE (%)** | | | | | | | | | | |
| **T** | 22 ± 10 | 24 ± 10 | 23 ± 8 | 28 ± 6 | 24 ± 8 | **21** ± 8 | **21** ± 13 | 25 ± 6 | 22 ± 14 | **21** ± 10 |
| **RH** | 50 ± 8 | 47 ± 7 | 32 ± 22 | 47 ± 20 | 32 ± 23 | 32 ± 20 | 33 ± 22 | 35 ± 13 | 35 ± 21 | **29** ± 17 |
| **P** | **21** ± 5 | 24 ± 9 | 31 ± 14 | 35 ± 16 | 30 ± 13 | 28 ± 12 | 27 ± 10 | 30 ± 13 | 27 ± 12 | 24 ± 10 |
| **U** | 55 ± 15 | 62 ± 16 | 40 ± 8 | 44 ± 9 | **38** ± 8 | **38** ± 9 | 43 ± 10 | 45 ± 13 | 44 ± 9 | 43 ± 10 |
| **K$_{in}$** | **17** ± 2 | **17** ± 3 | 26 ± 2 | 30 ± 11 | 26 ± 5 | 23 ± 4 | 29 ± 5 | 29 ± 9 | 27 ± 4 | 24 ± 2 |
| **L$_{in}$** | **18** ± 6 | **18** ± 6 | 30 ± 7 | 28 ± 5 | 32 ± 8 | 27 ± 6 | 31 ± 8 | 27 ± 7 | 30 ± 8 | 26 ± 8 |
| **Q$_H$** | 39 ± 13 | 41 ± 12 | 31 ± 8 | 35 ± 8 | **30** ± 7 | 31 ± 9 | 31 ± 9 | 35 ± 9 | 32 ± 9 | 34 ± 10 |
| **Q$_L$** | 25 ± 6 | 23 ± 8 | 23 ± 5 | 22 ± 6 | 24 ± 3 | 22 ± 6 | 21 ± 5 | 21 ± 6 | 22 ± 5 | **20** ± 6 |
| **Q$_M$** | **14** ± 4 | 15 ± 4 | 18 ± 3 | 23 ± 7 | 21 ± 4 | 19 ± 4 | 16 ± 3 | 22 ± 6 | 19 ± 3 | 18 ± 3 |

**Table S4:** Model performance, evaluated by $r_{sp}$ and NNSE, over the whole observational period in simulating daily: air temperature ($T$), relative humidity ($RH$), total precipitation ($P$), wind speed ($U$), incoming shortwave ($K_{in}$) and longwave ($L_{in}$) radiation, sensible ($Q_H$) and latent ($Q_L$) heat fluxes and total melt energy ($Q_M$). The melt energy is estimated according to the SEB model (Eq. 1). The WRF runs are based on three configurations of physics parameterizations: REF, minNRMSE and TOPSIS. The model performance is shown as the mean ($\pm$ one standard deviation) across the six study sites, with equal weighing of each site. Values in bold highlight the best performing model for the given variable. Values in purple highlight a statistically significant correlation at the 5 % confidence level for at least four of the six glacier sites.

| Variable | ERA5 30 km | ERA5-Land 9 km | WRF 3.3 km | | | WRF 1.1 km | | |
| --- | --- | --- | --- | --- | --- | --- | --- | --- |
| | | | REF | minNRMSE | TOPSIS | REF | TOPSIS | minNRMSE |
| **$r_{sp}$** | | | | | | | | |
| **T** | $0.86 \pm 0.19$ | $0.89 \pm 0.14$ | $0.89 \pm 0.05$ | $0.86 \pm 0.04$ | $0.88 \pm 0.03$ | $\mathbf{0.91} \pm 0.03$ | $0.88 \pm 0.03$ | $0.90 \pm 0.01$ |
| **RH** | $0.74 \pm 0.20$ | $\mathbf{0.75} \pm 0.15$ | $0.70 \pm 0.21$ | $0.55 \pm 0.19$ | $0.72 \pm 0.19$ | $0.71 \pm 0.20$ | $0.53 \pm 0.25$ | $0.71 \pm 0.17$ |
| **P** | $\mathbf{0.71} \pm 0.11$ | $\mathbf{0.71} \pm 0.11$ | $0.49 \pm 0.22$ | $0.44 \pm 0.28$ | $0.55 \pm 0.21$ | $0.51 \pm 0.24$ | $0.53 \pm 0.24$ | $0.49 \pm 0.26$ |
| **U** | $0.19 \pm 0.36$ | $0.13 \pm 0.32$ | $0.15 \pm 0.15$ | $0.10 \pm 0.16$ | $0.16 \pm 0.20$ | $\mathbf{0.23} \pm 0.20$ | $0.20 \pm 0.20$ | $0.19 \pm 0.24$ |
| **$K_{in}$** | $\mathbf{0.80} \pm 0.09$ | $\mathbf{0.80} \pm 0.09$ | $0.60 \pm 0.10$ | $0.58 \pm 0.19$ | $0.56 \pm 0.11$ | $0.52 \pm 0.16$ | $0.55 \pm 0.18$ | $0.47 \pm 0.14$ |
| **$L_{in}$** | $\mathbf{0.79} \pm 0.11$ | $0.78 \pm 0.12$ | $0.62 \pm 0.07$ | $0.56 \pm 0.11$ | $0.41 \pm 0.12$ | $0.54 \pm 0.07$ | $0.54 \pm 0.14$ | $0.35 \pm 0.17$ |
| **$Q_H$** | $0.29 \pm 0.36$ | $0.23 \pm 0.38$ | $0.54 \pm 0.23$ | $0.47 \pm 0.23$ | $\mathbf{0.50} \pm 0.24$ | $0.47 \pm 0.25$ | $0.49 \pm 0.33$ | $0.40 \pm 0.30$ |
| **$Q_L$** | $0.20 \pm 0.18$ | $0.15 \pm 0.22$ | $0.55 \pm 0.14$ | $0.48 \pm 0.19$ | $0.55 \pm 0.17$ | $0.59 \pm 0.12$ | $0.53 \pm 0.14$ | $\mathbf{0.61} \pm 0.10$ |
| **$Q_M$** | $\mathbf{0.86} \pm 0.03$ | $\mathbf{0.86} \pm 0.03$ | $0.71 \pm 0.08$ | $0.73 \pm 0.07$ | $0.69 \pm 0.12$ | $0.76 \pm 0.05$ | $0.75 \pm 0.07$ | $0.74 \pm 0.07$ |
| **NNSE (%)** | | | | | | | | |
| **T** | $58 \pm 25$ | $54 \pm 24$ | $56 \pm 18$ | $46 \pm 11$ | $55 \pm 18$ | $\mathbf{62} \pm 29$ | $51 \pm 11$ | $\mathbf{62} \pm 29$ |
| **RH** | $22 \pm 10$ | $24 \pm 11$ | $47 \pm 24$ | $30 \pm 23$ | $48 \pm 23$ | $\mathbf{48} \pm 24$ | $38 \pm 16$ | $44 \pm 23$ |
| **P** | $\mathbf{57} \pm 11$ | $52 \pm 19$ | $41 \pm 20$ | $36 \pm 18$ | $43 \pm 18$ | $45 \pm 16$ | $41 \pm 15$ | $46 \pm 20$ |
| **U** | $17 \pm 9$ | $14 \pm 6$ | $27 \pm 7$ | $23 \pm 9$ | $\mathbf{28} \pm 9$ | $24 \pm 7$ | $23 \pm 10$ | $23 \pm 7$ |
| **$K_{in}$** | $\mathbf{69} \pm 9$ | $68 \pm 10$ | $49 \pm 6$ | $44 \pm 20$ | $49 \pm 13$ | $43 \pm 4$ | $45 \pm 18$ | $47 \pm 10$ |
| **$L_{in}$** | $\mathbf{65} \pm 13$ | $\mathbf{65} \pm 14$ | $41 \pm 10$ | $44 \pm 10$ | $39 \pm 10$ | $39 \pm 10$ | $45 \pm 10$ | $41 \pm 11$ |
| **$Q_H$** | $30 \pm 12$ | $27 \pm 9$ | $39 \pm 10$ | $33 \pm 9$ | $\mathbf{41} \pm 10$ | $38 \pm 11$ | $33 \pm 10$ | $37 \pm 10$ |
| **$Q_L$** | $42 \pm 8$ | $45 \pm 10$ | $44 \pm 7$ | $47 \pm 8$ | $41 \pm 4$ | $\mathbf{50} \pm 7$ | $\mathbf{50} \pm 7$ | $47 \pm 5$ |
| **$Q_M$** | $\mathbf{74} \pm 10$ | $71 \pm 9$ | $62 \pm 6$ | $53 \pm 10$ | $56 \pm 5$ | $67 \pm 5$ | $54 \pm 11$ | $59 \pm 6$ |

**Table S5:** Model performance, evaluated by MAPE and NMBE, over the whole observational period in simulating daily: air temperature ($T$), relative humidity ($RH$), total precipitation ($P$), wind speed ($U$), incoming shortwave ($K_{in}$) and longwave ($L_{in}$) radiation, sensible ($Q_H$) and latent ($Q_L$) heat fluxes and total melt energy ($Q_M$). The melt energy is estimated according to the SEB model (Eq. 1). For evaluating $P$, only days with positive observed $P$ have been taken into account. The WRF runs are based on three configurations of physics parameterizations: REF, minNRMSE and TOPSIS. The model performance is shown as the mean ($\pm$ one standard deviation) across the six study sites, with equal weighing of each site. Values in bold highlight the best performing model for the given variable.

| Variable | ERA5 30 km | ERA5-Land 9 km | WRF 3.3 km | | | WRF 1.1 km | | |
|---|---|---|---|---|---|---|---|---|
| | | | REF | minNRMSE | TOPSIS | REF | minNRMSE | TOPSIS |
| **MAPE (%)** | | | | | | | | |
| T | 67 ± 60 | 71 ± 52 | 57 ± 43 | 67 ± 47 | **54 ± 36** | 63 ± 53 | 71 ± 53 | 53 ± 34 |
| RH | 33 ± 14 | 31 ± 13 | 16 ± 5 | 27 ± 12 | 16 ± 7 | **15 ± 5** | 20 ± 8 | 17 ± 6 |
| P | 287 ± 303 | **278 ± 299** | 438 ± 349 | 547 ± 465 | 471 ± 382 | 279 ± 215 | 325 ± 263 | 437 ± 335 |
| U | 61 ± 10 | 71 ± 7 | 43 ± 7 | 48 ± 9 | **41 ± 7** | 46 ± 13 | 49 ± 14 | 46 ± 11 |
| $K_{in}$ | **23 ± 7** | **23 ± 7** | 37 ± 7 | 32 ± 7 | 29 ± 4 | 41 ± 17 | 31 ± 4 | 32 ± 7 |
| $L_{in}$ | **4 ± 2** | 4 ± 3 | 7 ± 1 | 6 ± 2 | 7 ± 1 | 7 ± 2 | 6 ± 1 | 7 ± 1 |
| $Q_H$ | 94 ± 22 | 98 ± 18 | 89 ± 28 | 99 ± 31 | 87 ± 29 | 81 ± 23 | 101 ± 30 | **78 ± 25** |
| $Q_L$ | 279 ± 227 | **150 ± 76** | 268 ± 150 | 284 ± 185 | 255 ± 115 | 234 ± 191 | 263 ± 157 | 215 ± 155 |
| $Q_M$ | 35 ± 20 | **34 ± 20** | 48 ± 25 | 50 ± 26 | 50 ± 23 | 37 ± 16 | 51 ± 26 | 47 ± 27 |
| **NMBE (%)** | | | | | | | | |
| T | -9 ± 73 | 2 ± 69 | -13 ± 42 | -33 ± 50 | **0 ± 46** | -17 ± 48 | -43 ± 43 | -2 ± 37 |
| RH | 33 ± 14 | 31 ± 13 | **-3 ± 14** | 19 ± 23 | -6 ± 13 | -9± 11 | 11± 17 | -12 ± 9 |
| P | 235 ± 318 | 227 ± 317 | 352 ± 353 | 463 ± 473 | 368 ± 373 | **117 ± 206** | 231 ± 263 | 320 ± 335 |
| U | -61 ± 10 | -71 ± 7 | -20 ± 18 | -36 ± 14 | **-16 ± 20** | -40 ± 16 | -42 ± 13 | -38 ± 15 |
| $K_{in}$ | 18 ± 10 | 18 ± 10 | 20 ± 10 | -14 ± 19 | **0 ± 11** | 31 ± 24 | -7 ± 21 | 9 ± 14 |
| $L_{in}$ | **0 ± 4** | **0 ± 4** | -5 ± 3 | **0 ± 4** | -2 ± 3 | -5 ± 3 | -1 ± 3 | -2 ± 2 |
| $Q_H$ | -86 ± 32 | -96 ± 19 | -36 ± 28 | -68 ± 13 | **-17 ± 28** | -59 ± 27 | -77 ± 14 | -44 ± 23 |
| $Q_L$ | -194 ± 237 | -126 ± 78 | 41 ± 150 | -70 ± 83 | **2 ± 140** | 5 ± 195 | -5 ± 137 | -24 ± 151 |
| $Q_M$ | -6 ± 9 | -13 ± 9 | **-4 ± 26** | -32 ± 29 | -6 ± 25 | -9 ± 19 | -28 ± 28 | -14 ± 19 |

**Minor Comments**

*R2 C2:* Page 1, line 22: I suggest rephrasing to "…and are increasingly losing a considerable amount of mass ….."

Done.

*R2 C3:* Page 2, line 39: Why does SEB models not require precipitation?

Please see our comment to R2 C1 regarding the importance of rain heat flux in seasonal SEB. We now better clarify why precipitation is not considered in the SEB model.

*R2 C4:* Page2, line 43: I suggest rephrasing to " …fewer than 100 sites worldwide, and only a handful in Western Canada…."

Done.

*R2 C5:* Page 3, section starting on line 62. Here I suggest including a citation of the recent work by Eidhammer et al 2021 (https://hess.copernicus.org/articles/25/4275/2021/), where they use the detailed snow model Crocus within the WRF-Hydro model to estimate glacier melt (and streamflow). They used 1 km downscaled WRF simulations over a glacier in Norway for four seasons.

This reference is now incorporated in the text (lines 83-86):

*More recently, Eidhammer et al. (2021) used WRF downscaling to 1 km grid spacing coupled with snow-pack modeling through the WRF-Hydro model (Gochis et al., 2020), showing a good agreement between the WRF output and in-situ meteorological observations at a glacier in Norway over four years.*

*R2 C6:* Page 3: In the discussion of using dynamical downscaling, you might want to add a comment related to the paper by Lundquist et al. 2019 with the title: "Our Skill in Modeling Mountain Rain and Snow is Bypassing the Skill of Our Observational Networks". https://doi.org/10.1175/BAMS-D-19-0001.1. I think that this can add to the argument in this manuscript to use downscaling for SEB modeling.

This reference is now incorporated in the text (lines 56-60):

*An alternative to statistical is dynamical downscaling: a physics-based approach that utilizes a regional climate model (RCM), nested within a reanalysis or global climate model, to compute meteorological fields at a desirable spatial resolution, often <10 km. A well-configured high-resolution RCM outperforms radar and satellite-derived estimates of total annual rain and snowfall within mountainous regions (Lundquist et al., 2019).*

*R2 C7:* Page 3, line 22. I suggest adding a citation to the paper by Liu et al 2011 "High-Resolution Simulations of Wintertime Precipitation in the Colorado Headwaters Region: Sensitivity to Physics Parameterizations" (https://doi.org/10.1175/MWR-D-11-00009.1). They tested several different WRF physics parameterizations over the Colorado headwaters region.

This reference is now incorporated in the text (lines 93-95):

*A relatively underexplored limitation in using WRF in glacier studies is the model's potentially large sensitivity to the choice of physics parameterization schemes as noted in many non-glacier studies (e.g., Liu et al., 2011; Zeyaeyan et al., 2017; Gbode et al., 2019; Pervin and Gan, 2020; Shirai et al., 2022).*

*R2 C8:* Page 3, line 94. The Thompson-Eidhammer scheme (https://doi.org/10.1175/JAS-D-13- 0305.1) has also been used for Glacier studies

We have incorporated the Thompson-Eidhammer scheme in the text (lines 210-215). We now moved the description of previously used parameterization schemes to the "Data and Methods" section as per suggestion of referee #1 (R1 C7).

*Lines 210-215: For example, the most commonly used schemes in glaicer studies include RRTMG (Iacono et al., 2008), CAM (Collins et al., 2004), Dhudia (Dudhia, 1989) and Goddard (Max and Suarez, 1994; Matsui et al., 2018) for radiation, the Grell 3D Ensemble (Grell, 1993; Grell and Dévényi, 2002), the Kain-Fritsch (Kain, 2004) and the Betts-Miller-Janjić (Janjić, 1994) schemes for cumulus convection, and the Morrison two-moment (Morrison et al., 2009), the Thompson (Thompson et al., 2008) and the updated aerosol–aware Thompson-Eidhammer (Thompson and Eidhammer, 2014) schemes for microphysics.*

*R2 C9:* Page 5, Table 5, caption: What is meant by "full days"?

The table caption is rephrased to:

*Table. 1. Characteristics of the study sites. Only days with 24-hour observations have been taken into account for the observational periods.*

*R2 C10:* Page 5, line 147: I suggest rephrasing: "…the accumulation zone of the Conrad glacier in 2016."

Done.

*R2 C11:* Page 6, line 163: The way I read the sentence, the reference to Table 1 indicates that there is some information in regards to the melting surface with intermittent fresh snowfall in the Table 1. I do not think the reference to Table 1 is necessary here.

The reference to Table 1 is now removed in this line and the following lines (159-171).

*R2 C12:* Page 8, lines 2 and 3: I suggest to clarify that both 3D and 2D ERA 5 fields (I assume some 2D fields are used at initialization) are used as forcing data for the WRF model.

We revised the text (lines 179-180):

*Hourly two- and three-dimensional ERA5 reanalysis data is also used to provide initial and lateral boundary conditions to the WRF model.*

*R2 C13:* Page 8, line 193. The way I read this line, the d1 domain for all the 4 glaciers are the same. However, Figure 3 shows that d1 is different between Kaskawulsh and the other glaciers. Can you please clarify?

The domain d1 does differ between Kaskawulsh and other glaciers. Thank you for spotting this discrepancy. We revised the sentence (lines 186-190):

*We ran the WRF model, version 4.1.3, configured with four nested domains of 30 km (d1), 10 km (d2), 3.3 km (d3) and 1.1 km (d4) horizontal grid spacing, with the parent domain (d1) covering the bulk of North America and the North-East section of the Pacific Ocean (Figure 3). The domains d1 and d2 are kept the same for the three glaciers in the interior of British Columbia (Castle Creek glacier, Nordic glacier and Conrad glacier), while d3 and d4 are set differently for each of the three glaciers in order to be centered at the AWS location.*

*R2 C14:* Page 8, line 198. I assume that you mean that many physics variables are updated every 2.2 s, not outputted? And most likely, the radiation and land surface variables are probably not calculated every 2.2 seconds, but perhaps somewhere between every 5 or 30 minutes? Also make sure if the hourly outputs indeed are hourly averages. Typically, most of the WRF outputs are instantaneously outputs, with some of them being accumulated.

Yes, there are 30 minutes between radiation physics calls in our model, as recommended for our grid spacing of 30 km for the outer domain. WRF updates many physics variables every 2.2 s in the inner-most domain and, as we use WRF "tslist" (time series) output, we get WRF output for every time step

(i.e., 2.2 s for the inner-most domain). We average this time series output to hourly and daily averages. The text is revised accordingly (lines 194-195):

*We use a time step of 2.2 s for the most inner domain and save the selected set of variables as hourly and daily averages.*

*R2 C15:* Page 8, line 199: Table 2 does not describe any of the output saved. I suggest remove the reference to Table 2.

We removed the reference to Table 2.

*R2 C16:* Page 9, line 208. I wonder if it would be helpful to add a delta elevation from AWS in table S1 as well. In this way, it would be easier to see the actual elevation difference.

The delta elevation for each of the input datasets (ERA5, ERA5-Land, WRF at 3.3 km, WRf at 1.1 km, SRTM/ASTER) in comparison to the AWS elevation is now added in Table S1.

**Table S1:** Elevation for each study site (in meter above sea level) derived from AWS (on-site GPS), ERA5 at 30 km grid spacing, ERA5-Land at 9 km, WRF at 3.3 km and 1.1 km, and a high-resolution DEM at 30 m grid spacing: SRTM (NASA JPL, 2013; Farr et al., 2007) for Castle Creek, Nordic and Conrad glaciers, and ASTER (ASTER, 2019; Abrams et al., 2020) for Kaskawulsh glacier). Numbers in brackets show the respective difference to the AWS elevation.

| Glacier site | AWS | ERA5 | ERA5-Land | WRF 3.3 km | WRF 1.1 km | SRTM / ASTER |
|---|---|---|---|---|---|---|
| Castle Creek 2012 | 1967 | 1762 (-205) | 1987 (+20) | 2157 (+190) | 1915 (-52) | 1977 (+10) |
| Nordic 2014 | 2208 | 1785 (-423) | 1866 (-342) | 2124 (-84) | 2298 (+90) | 2203 (-5) |
| Conrad 2015 | 2138 | 1901 (-237) | 2145 (+7) | 2412 (+274) | 2184 (+46) | 2163 (+25) |
| Conrad 2016 AWS$_1$ | 2164 | 1901 (-263) | 2145 (-19) | 2567 (+403) | 2217 (+53) | 2182 (+18) |
| Conrad 2016 AWS$_2$ | 2909 | 1901 (-1008) | 2145 (-764) | 2618 (-291) | 2944 (+35) | 2910 (+1) |
| Kaskawulsh 2019 | 1666 | 2122 (+456) | 2159 (+493) | 1709 (+43) | 1659 (-7) | 1709 (+43) |

*R2 C17:* Page 9, line 214. On page 3, it is stated that the Microphysics by Morrison is the most commonly used in glacier studies, but in this work, the Thompson microphysics is used. Please clarify this discrepancy. Also see page 27, line 597

Yes, the referee is correct. Both schemes have been previously used in glacier studies. The list of physics parameterization schemes for our sensitivity analysis in Table S2 now matches with the schemes described in the text (lines 209-221). During the sensitivity analysis, we tested both the Morrison 2-moment and the Thompson schemes and found that the Thompson scheme showed better performance in overall melt energy at our glacier sites.

*Lines 209-221: The WRF model comes with various options for physics parameterizations (Skamarock et al., 2019), but previous glacier studies with WRF have used some parameterization schemes more often than others. For example, the most commonly used schemes in glaicer studies include RRTMG (Iacono et al., 2008), CAM (Collins et al., 2004), Dhudia (Dudhia, 1989) and Goddard (Max and Suarez, 1994; Matsui et al., 2018) for radiation, the Grell 3D Ensemble (Grell, 1993; Grell and Dévényi, 2002), the Kain-Fritsch (Kain, 2004) and the Betts-Miller-Janjić (Janjić, 1994) schemes for cumulus convection, and the Morrison two-moment (Morrison et al., 2009), the Thompson (Thompson et al., 2008) and the updated aerosol–aware Thompson-Eidhammer (Thompson and Eidhammer, 2014) schemes for microphysics. The local-closure Mellor–Yamada–Nakanishi–Niino (MYNN) level 2.5 (Nakanishi and Niino,*

*2006, 2009; Olson et al., 2019) and Mellor-Yamada-Janjic (Janjić, 1994; Mesinger, 1993) schemes, as well as the non-local closure Yonsei University (Hong et al., 2006) scheme have been most commonly used for boundary layer, and the revised MM5 (Jiménez et al., 2012) and Eta Similarity (Monin and Obukhov, 1954; Janjić, 1994, 1996, 2002) schemes for surface layer. The Noah (Tewari et al., 2004) and Noah-MP (Niu et al., 2011; Yang et al., 2011) land surface models are most commonly used in glacier studies, but the WRF simulations over non-glacierized terrain are shown to vary substantially depending on which of the two land surface models is used (Milovac et al., 2016).*

*R2 C18:* Page 13, line 277: It is stated that the goal is to evaluate daily timeseries of simulated energy available for melt. As shown in Fitzpatrick 2017, the QR was shown to have considerable influence on SEB when considering daily and sub-daily timescales. I am wondering if ignoring QR in this study is then valid?

We have revised the sentence (lines 291-296) to align more consistently with the stated objectives in the Introduction. Our primary focus is to assess the performance of ERA5 and WRF in simulating seasonal melt energy for long-term glacier melt simulations. Additionally, we explore the daily-scale performance by analyzing the timeseries of daily fluxes rather than solely relying on net (average) fluxes across the observational period.

As mentioned earlier in comment R2 C1, over a melting season, rainfall can significantly contribute (up to 20%) to daily melt energy, as demonstrated by Fitzpatrick et al. (2017). However, the uncertainty in modeling rain heat flux is relatively large and might potentially be grounded in unsupported assumptions (Hock, 2005; Fitzpatrick et al., 2017). Despite these significant contributions on daily scales, the low occurrence and duration of these extreme precipitation events does lead to a contribution of <1% to the total melt energy over the melting season, as shown by Fitzpatrick et al. (2017).

We chose not to include the modeled rain heat flux in our SEB model, as we are focused only on the key contributors to melt energy on seasonal scales. Nonetheless, we have now included a comparison between modeled (WRF/ERA5) and observed precipitation at our sites, considering the relevance of fresh snowfall events for albedo (refer to lines 355-362 and 374-379, as well as Figures S2 and S3; see R2 C1).

*R2 C19:* Page 13, line 286. The study by Eidhammer et al. 2021 also shows that the albedo, when using Noah-MP does not perform well over glaciers (especially in the ablation region).

Thank you for pointing out this study. The reference is now incorporated (lines 303-304):

*The discrepancy between WRF and observed albedo at glaciers, especially in the ablation zone, has also been noted previously in Eidhammer et al. (2021).*

*R2 C20:* Page 18, line 371: Please specify that this is the WRF REF data.

Done.

*R2 C21:* Page 19, section 3.3 I think that this section should come before section 3.2, since you are using the results from section 3.3 in describing results in section 3.2.

Done.

*R2 C22:* Page 19, line 404: Level 3, not Level 3

Done.

*R2 C23:* Page 23, line 490: Did you consider employing different roughness lengths over snow versus ice?

Roughness lengths do differ between snow and ice surfaces, and for most of our sites they are different by two orders of magnitude $(z_{0v} \sim 10^{-3} \text{m} \ z_{0T} \sim z_{0q} \sim 10^{-5} \text{m})$. We now make this more clear in the text (lines 283-290, Table S3):

*We use constant roughness lengths for each site, which have been adopted from the EC-derived turbu-lent fluxes in previous studies at these glacier sites (Table S3; Radić et al., 2017; Fitzpatrick et al., 2017, 2019; Lord-May and Radić, 2023), while for the stability corrections, based on the assessed stability conditions, we use the functions applied previously in Fitzpatrick et al. (2017). The order of magnitude in these EC-derived values for roughness lengths* $(z_{0v} = 10^{-3} \text{m} \ z_{0T} = z_{0q} = 10^{-5} \text{m}).)$ *agree with commonly assumed values for glaciers in mid-latitudes (Hock, 2005). Vapor pressure $e_z$ at height z is calculated from the relative humidity RH at height z using the August-Roche-Magnus formula (Alduchov and Eskridge, 1997). For more details on the bulk method and the stability corrections used in the study, readers are referred to Fitzpatrick et al. (2017).*

---

## Author Response (AR2)

**Responses to Technical Corrections**

Dear Dr. Collier,

Thank you for accepting our manuscript for publication in TC subject to addressing technical corrections. We appreciate your diligent handling of the paper and the efforts of the reviewers in providing valuable feedback. We have addressed all the suggestions, questions, and corrections in this final version. Please find a detailed response to the comments below, with the comments in black and our corresponding replies in blue.

Thank you again for your time and effort.

Sincerely,
Christina Draeger on behalf of all coauthors
* * *
Thank you for your revised manuscript and responses, as well as for your patience with the editorial decision. I am pleased to accept your manuscript for publication in TC subject to addressing the following technical corrections:

1. Line 38: can you add a reference for process-based models being more reliable? Otherwise, consider rephrasing along the lines of these models not relying on the temporal stationarity of melt factors.

Rephrased to:

*Line 37: Since these models capture the physical processes that are happening at the glacier surface, they do not rely on the temporal stationarity of melt factors, as is the case in temperature-index models. However, they require a larger number of input variables, including incoming shortwave and longwave radiation, temperature, relative humidity, wind speed and precipitation.*

2. Line 90: consider rephrasing to "on the order of a kilometre (e.g., Erler...)" as there are decadal applications of WRF over mountainous regions at less than 4-km grid spacing in the convection-permitting climate modelling literature.

Rephrased accordingly.

3. Consider making the REF namelist available via open repository so that the workflow and model configuration (for example, in which domains terrain shading is applied) are reproducible.

The REF namelist will be made available.

4. From my memory, the glacier subroutines in Noah and Noah-MP did not differ significantly. To help interpret the sensitivity results, consider adding a sentence to the methods about the difference in how glacier surfaces are treated between the two LSMs. Can you also please explicitly state if the albedo parameters were only adjusted in the Noah-MP (line 442) but not in the Noah LSM?

Added:

*Line 221: Noah-MP, which is a more sophisticated version of Unified Noah, includes multiple snow layers, representing percolation, retention, and refreezing of meltwater within the snowpack rather than in the snow–atmosphere and snow–soil interface as is the case with Unified Noah (Suzuki and Zupanski, 2018).*

*Line 457: In both land surface models, the glacier surface albedo is calculated as a weighted average of land ice albedo and snow albedo based on snow cover fraction (He et al., 2023).*

*Line 450: No changes were applied to the albedo representations within Unified Noah.*

5. Could you provide a justification for using the BMJ cumulus scheme in all domains, even the km-scale ones? Does this scheme have a scale-aware component?

Added:

*Line 536: We note that none of the cumulus schemes used in this study is scale-aware. Theoretically, cumulus parameterizations are only valid for coarse spatial grids of more than 10 km in order to release latent heat in the convective columns (Zhang et al., 2012). The parameterizations can also help to trigger mesoscale convection (5–10 km). For a grid spacing of 3–5 km or less, it is recommended to switch "off" the cumulus schemes as the model can explicitly resolve deep convection and simulate convective storms (Skamarock and Klemp, 2008). However, it has also been recommended to keep this parameterization "on" for grid spacing of 1–10 km to avoid accumulated energy at grid points (Gerard, 2007). The cumulus parameterization scheme has been consistently turned "off" below 3 km in previous glacier studies (e.g., Mölg and Kaser, 2011; Collier et al., 2013, 2015; Aas et al., 2016). Between 3 and 5 km, some studies used the cumulus parameterization scheme (e.g., Mölg and Kaser, 2011), while others explicitly resolved deep convection without parameterization (e.g., Aas et al., 2016).*

6. Line 304: Unrepresentative surface albedo in the ablation zone was also demonstrated by Collier et al. (2013). It would be worth mentioning (unless I missed it) that although observed albedo is used in the SEB model, the (poorly) simulated albedo will still impact the near-surface meteorological forcing fields.

*Rephrased to:*

*Line 306: The discrepancy between WRF and the observed albedo on glaciers, especially in the ablation zone, has also been noted in previous glacier studies (Collier et al., 2013; Eidhammer et al., 2013)*

*Added:*

*Line 309: We note that while we incorporate observed albedo into the SEB model, the inaccurately simulated albedo in WRF still influences the near-surface meteorological forcing fields.*

Thank you for your efforts and congratulations on an interesting contribution to the glacier SEB & MB modelling community!

Best regards,
Emily Collier